# MC-DiT: Contextual Enhancement via Clean-to-Clean Reconstruction for Masked Diffusion Models

**Guanghao Zheng, Yuchen Liu, Wenrui Dai** [*]**, Chenglin Li, Junni Zou, Hongkai Xiong**
School of Electronic Information and Electrical Engineering
Shanghai Jiao Tong University

## Abstract

Diffusion Transformer (DiT) is emerging as a cutting-edge trend in the landscape of generative diffusion models for image generation. Recently, masked-reconstruction strategies have been considered to improve the efficiency and semantic consistency in training DiT but suffer from deficiency in contextual information extraction. In this paper, we provide a new insight to reveal that noisy-to-noisy masked-reconstruction harms sufficient utilization of contextual information. We further demonstrate the insight with theoretical analysis and empirical study on the mutual information between unmasked and masked patches. Guided by such insight, we propose a novel training paradigm named **MC-DiT** for fully learning contextual information via diffusion denoising at different noise variances with clean-to-clean mask-reconstruction. Moreover, to avoid model collapse, we design two complementary branches of DiT decoders for enhancing the use of noisy patches and mitigating excessive reliance on clean patches in reconstruction. Extensive experimental results on $256{\times}256$ and $512{\times}512$ image generation on the ImageNet dataset demonstrate that the proposed MC-DiT achieves state-of-the-art performance in unconditional and conditional image generation with enhanced convergence speed.

## 1 Introduction

Diffusion Probabilistic Models (DPMs) [19, 29, 42, 43] have emerged as front-runners in the latest advancements of image-level generative models, and surpass previous state-of-the-art generative models [10, 14]. DPMs corrupt an input image into a noise obeying the normal distribution by gradually injecting Gaussian noise and recover the image from the noise with step-by-step denoising via a pretrained network [42, 43]. U-Net [38] is popular in early works [37, 35] to predict noise from disrupted images for image generation. Recently, Diffusion Transformer (DiT) [34] has been widely considered for DPMs [19, 29, 42, 43] in conditional image generation [4, 37], video generation [16, 22, 30], and 3D generation [15, 25, 36] due to its excellent scalability and superior performance.

Different from vision transformers (ViTs) [11], DiT is trained to predict Gaussian noise from disrupted images at different noise levels. To achieve large-scale training, DiT suffers from slow convergence and heavy computational burden in the training process [49]. Moreover, due to its goal of noise prediction, DiT causes semantic inconsistency in generated images, since it struggles to learn contextual information in different regions of images for noise prediction [13]. To solve these problems, mask-reconstruction is introduced into the denoising-based training schedule for DiT [13, 49, 48]. Inspired by masked autoencoder (MAE) [18], DiT is trained to predict masked noisy patches from the given unmasked noisy patches. MDT [13] pioneers to propose the asymmetrical diffusion transformer that integrates mask-reconstruction with denoising, which employs encoders to extract features from unmasked noisy patches and a lightweight decoder to reconstruct masked patches via extracted features. MaskDiT [48] accelerates the training process by masking at most 50%

---

[*]Correspondence to Wenrui Dai <daiwenrui@sjtu.edu.cn>.

38th Conference on Neural Information Processing Systems (NeurIPS 2024).

noisy image patches. SD-DiT [49] introduces self-supervised discriminative objective for knowledge distillation to reduce the fuzzy relation between the mask-reconstruction and denoising. Despite superior performance over vanilla DiT, they are deficient in exploiting contextual information by neglecting different noise scales in different steps of diffusion process.

In this paper, we revisit mask-reconstruction in training DiT and reveal that **reconstructing masked noisy patches from unmasked noisy patches harms contextual information extraction**. This issue is exaggerated under large noise variances, since both unmasked and masked noisy patches are similar to standard Gaussian noise and contain little contextual information. We demonstrate this phenomenon in Figure 1(a) by evaluating the mutual information between unmasked output patches and masked patches at different noise variances for different methods. With the growth of noise variance, mutual information in noisy image patches generated by MDT [13] and MaskDiT [48] decreases sharply, while mutual information in vanilla noisy images decreases slowly. This fact suggests that the information in masked patches rarely comes from unmasked patches, and thereby, the contextual information is not sufficiently exploited. We further demonstrate this empirical finding with theoretical analysis on mutual information and the mask graph [46], as elaborated in Propositions 2 and 3.

To address this problem, we propose MC-DiT to reconstruct clean unmasked patches from clean masked patches rather than resort to noisy patches. Benefiting from the merit that clean-to-clean reconstruction is not influenced by the noise, the proposed MC-DiT is able to learn contextual information via the diffusion denoising process at all noise scales. Furthermore, to avoid model collapse caused by over-emphasizing clean patches but neglecting noisy patches, we design two complementary branches to enforce the model focusing more on denoising. In summary, our contributions are listed as below.

- We provide a new insight that noisy-to-noisy mask-reconstruction is insufficient in extracting contextual information. We demonstrate the insight on mask-reconstruction with theoretical analysis and empirical study on mutual information between unmasked and masked patches.
- We propose MC-DiT, a novel training paradigm with new mask-reconstruction objective, to fully exploit contextual information with clean-to-clean reconstruction. We further design two complementary branches of DiT decoders to avoid model collapse and focus on denoising.
- We evaluate the proposed MC-DiT in 256×256 and 512×512 image generation on the ImageNet dataset and achieve state-of-the-art FID score for DiT backbones of various scales.

## 2    Related Work

**Diffusion Probabilistic Models.** Diffusion Probabilistic Models (DPMs) [19, 42, 43] have attracted increasing attention due to their superior image generation ability compared with preceding generative models [14, 47]. Denoising diffusion probabilistic model (DDPM) [19] significantly advances generative models, particularly in tasks such as text-guided image synthesis. In specific, DDPM is realized as a Markov chain [32] that contains forward process and reverse process. In the forward process, clean images are disrupted by Gaussian noise step by step, and in the reverse process, the images are generated from the Gaussian noise with step-by-step denoising. The commonly used U-Net model [38] is trained to predict the Gaussian noise from noisy images. Score matching method [43] is introduced into diffusion models to design the forward and reverse process with elegant Stochastic Differential Equation (SDE) [44]. EDM [21] analyzes the design space of diffusion models and disentangles the effects of parametrization, sampling, and training. To address the time-consuming iterative issue inherent in DPMs, several methods apply fast sampling strategy [27, 28] or latent diffusion training strategy [37].

**Transformers in Diffusion Models.** In DPMs, the most commonly used architecture for noise prediction is U-Net [38], which is a symmetric encoder-decoder framework. Recently, transformers provide a new perspective to noise prediction due to their excellent multi-modality fusion ability and remarkable scaling properties. U-ViT [3] integrates time embedding, image patches, and conditional patches into overall tokens and applies residual connection into transformers for consistency in generation. DiT [34] claims that transformers applied in DPMs realize superior performance and achieve the scaling law. Therefore, various works adopt and improve DiT into 2D image generation [34, 35], video generation [30, 16, 22], and 3D generation [15, 25, 36]. In this paper, we take the DiT as our backbone and change the input from noisy patches to clean patches.

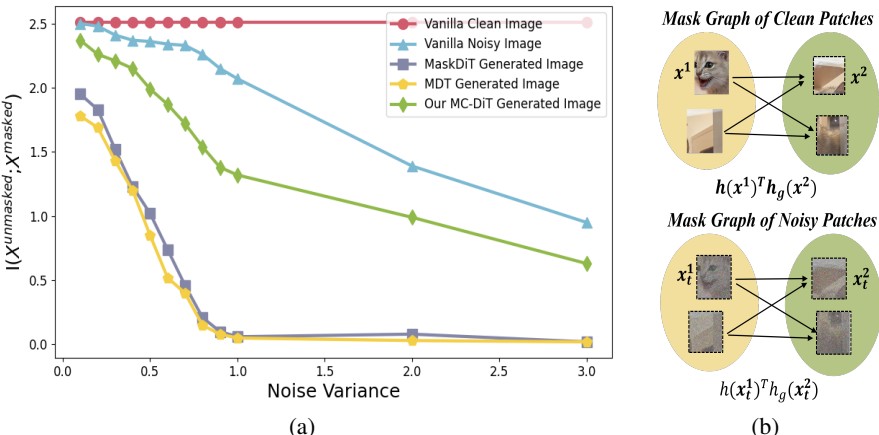

(a)                                                                     (b)

Figure 1: (a) Mutual information of different methods between generated masked patches and unmasked patches. We generate masked noisy patches from unmasked noisy patches and calculate mutual information $I(X^{unmasked}; X^{masked})$ under various noise scales. The 'vanilla clean images' and 'vanilla noisy images' denote the real clean/noisy images, which are the upper bound of the mutual information. The other lines are computed with the generated images by three strategies. (b) Mask graph [46] in different reconstruction targets. The left yellow ellipse denotes unmasked patches and the right green one denotes masked patches. The black arrow denotes the positive pairs to pull in.

**Masked Training with Transformers.** Mask-reconstruction has been broadly applied into convolutional networks and transformers [11]. MAE [18] takes the mask-reconstruction to pretrain transformers and achieves stunning contextual modeling capabilities and zero-shot performance. Inspired by this method, various strategies for masked training have been introduced into transformers. For example, ConvMAE [12] leverages masked convolution to prevent information leakage. FreMiM [45] converts images into frequency domain and applies masked training for frequency information reconstruction. MultiMAE [1] utilizes masked training into multi-modality inputs for cross-modal fusion and generation. SdAE [8] improves masked autoencoder via self distillation. In summary, the mask-reconstruction training objective transfers the information from unmasked patches to masked patches, and thereby, enhances contextual semantic modeling ability.

## 3 Proposed Method

### 3.1 Preliminaries

**Masked AutoEncoders [18].** Masked AutoEncoder (MAE) is a significant unsupervised pretraining paradigm in computer vision, which reconstructs masked patches from unmasked patches. Given an image $x$, MAE first patchifies it into $N$ patches denoted by $\tilde{x} \in R^{N \times c}$, where $c$ is the channel dimension. A random binary mask $m \in \{0,1\}^N$ is applied to obtain masked patches $x_1 = x[1-m] \in R^{N_1 \times c}$ and unmasked patches $x_2 = x[m] \in R^{N_2 \times c}$. $N_1$ and $N_2$ are the number of masked and unmasked patches. An encoder-decoder framework $h = g \circ f$ is then applied. The encoder $f$ maps the unmasked patches $x_1$ into latent space $z_1 = f(x_1)$, while the decoder reconstructs the pixel value of masked patches $x'_2 = g(z_1)$. The MAE is trained to ensure the reconstruction ability by minimizing the Mean Square Error (MSE) loss $\mathcal{L}_{MAE} = \mathbb{E}_x \mathbb{E}_{x_1, x_2 | x} \|g(f(x_1)) - x_2\|^2$. U-MAE [46] provides a theoretical understanding of MAE and establishes connection between MAE and contrastive learning [33, 17, 7].

**Proposition 1 ([46])** *The lower bound of the MAE loss is*

$$\mathcal{L}_{\mathrm{MAE}} \geq -\mathbb{E}_{x_1, x_2} h(x_1)^T h_g(x_2) - \varepsilon + \mathrm{const} = \mathcal{L}_{\mathrm{asym}} - \varepsilon + \mathrm{const}, \tag{1}$$

where $\mathcal{L}_{\mathrm{asym}}$ denotes the asymmetric alignment loss in [46], $\varepsilon$ is the fitting error, and $h_g = g \circ f_g$ is the pseudo autoencoder that satisfies $\mathbb{E}_x \|h_g(x_2) - x_2\|^2 \leq \varepsilon$.

U-MAE [46] combines Proposition 1 with the mask graph in Figure 1(b) (upper) to represent the contrastive objective in MAE. Specifically, Proposition 1 demonstrates that the MAE loss is equal

to the contrastive loss $\mathcal{L}_{asym}$, which calculates the similarity of $h(x_1)$ and $h_g(x_2)$. If $x_1$ and $x_2$ are neighboring patches, $\mathcal{L}_{asym}$ is minimized when positive pairs (*i.e.*, $x_1$ and $x_2$) are pulled closer. Proposition 1 is consistent with the mask graph in Figure 1 (b), where unmasked and masked patches are considered as contrastive pairs. Thus, we employ mask graph as an effective tool to analyze the contrastive objective of MAE.

**Diffusion Probabilistic Models [19, 42, 43].** Diffusion Probabilistic Models (DPMs) emulate the diffusion process of physical atoms to convert the standard Gaussian distribution into the target distribution via differential equations. In the forward process, clean data $x_0 \sim P_{data}(x_0)$ is corrupted into Gaussian noise $x_T \sim \mathcal{N}(0, \sigma_{max}^2 I)$ step by step via stochastic differential equation (SDE):

$$\mathrm{d}x = f(x_t, t)\mathrm{d}t + g(t)\mathrm{d}w, \tag{2}$$

where $f$ is the drift coefficient, $g$ is the diffusion coefficient, $w$ is a standard Wiener process, and $t$ is the time value from 0 to $T$. The reverse process generates target samples using the inverse SDE:

$$\mathrm{d}x = [f(x, t) - g(t)^2 \nabla_x \log p_t(x)]\mathrm{d}t + g(t)\mathrm{d}\tilde{w}, \tag{3}$$

where $\tilde{w}$ is a reverse-time Wiener process. Following EDM [21] to set $f(x, t) = 0$ and $g(t) = \sqrt{2}t$, the forward and reverse SDEs are reformulated as $\mathrm{d}x = \sqrt{2}t\mathrm{d}w$ and $\mathrm{d}x = -t\nabla_x \log p_t(x)\mathrm{d}t$, where $s(x, t) = \nabla_x \log p_t(x)$ is the score function. To solve the reverse-time SDE, a denoising network $D_\theta(x, t)$ is trained to minimize the score matching loss:

$$\mathbb{E}_{x_0 \sim p_{data}(x_0)}\mathbb{E}_{n \sim \mathcal{N}(0, t^2 I)}\|D_\theta(x_0 + n, t) - x_0\|^2. \tag{4}$$

As a result, the score function is estimated by $s(x, t) = (D_\theta(x, t) - x)/t^2$. During training, at step $t$, noisy images $x_0 + n$ are sent to the denoising network $D_\theta(x, t)$ to predict the clean images $x_0$. However, directly optimizing this objective leads to poor contextual information [13]. To solve this problem, the mask-reconstruction is applied into denoising network [13, 48, 49].

### 3.2 Contextual Information in Noisy Patches Reconstruction

We first review the mask-reconstruction between noisy patches and point out that applying noisy patches reconstruction task into the training process of DiT is ineffective and leads to insufficient contextual information utilization. With the mutual information and mask graph, we propose two propositions to demonstrate this claim, where the first is for the mutual information of input-output patches and the second is for the contrastive objective of input unmasked-masked patches.

**Mutual information between input and output patches.** Besides the mutual information between masked and unmasked output patches in Figure 1(a), we consider the mutual information between input noisy patches and output (noisy and clean) patches. MaskDiT [48] and MDT [13] reconstruct masked noisy output patches from input unmasked noisy input patches, whereas SD-DiT [49] recovers unmasked clean output patches from masked noisy input patches, as elaborated below.

- **MaskDiT & MDT.** Noisy patches $x_t$ at step $t$ are obtained by injecting noise $n \sim \mathcal{N}(0, t^2 I)$ into clean patches $x_0$. Masked and unmasked noisy patches are generated from $x_t$ by $x_t^1 = x_t[m]$ and $x_t^2 = x_t[1 - m]$ using a random binary mask $m$. $x_t^2$ is reconstructed from $x_t^1$ by minimizing the MAE loss $\mathcal{L}_{\text{Mask-MAE}} = \mathbb{E}_{x_t}\mathbb{E}_{x_t^1, x_t^2|x_t}\|g(f(x_t^1)) - x_t^2\|^2$ for the encoder $f$ and decoder $g$. The ability to exploit contextual information is measured by mutual information $\mathcal{I}(x_t^1; x_t^2)$.
- **SD-DiT.** SD-DiT extracts latent features of $x_t^1$ and concatenates them with masked noisy patches $x_t^2$ to predict clean patches $x_0^1$. The MAE loss $\mathcal{L}_{\text{SD-MAE}} = \mathbb{E}_{x_t}\mathbb{E}_{x_t^1, x_t^2|x_t}\|g(f(x_t^1), x_t^2) - x_0^1\|^2$. The ability to exploit contextual information is measured by mutual information $\mathcal{I}(x_0^1; x_t^2)$.

The contextual information in both two formulations is transferred from the noisy patches to noisy patches. [2] Subsequently, we analyze the contextual information utilization ability of mask-reconstruction via calculating mutual information of masked and unmasked patches.

---

[2]Although the reconstruction targets of SD-DiT are clean patches, it is equivalent to distinguish the $x_0^1$ and noise $n$. Therefore, the contextual information is used for better prediction of $n$.

**Proposition 2** *Given masked and unmasked clean patches $x_0^1$ and $x_0^2$ and their noisy versions $x_t^1$ and $x_t^2$, the mutual information $\mathcal{I}(x_t^1; x_t^2)$, $\mathcal{I}(x_0^1; x_t^2)$, and $\mathcal{I}(x_0^1; x_0^2)$ satisfy that*

$$\mathcal{I}(x_0^1; x_t^2) \approx \mathcal{I}(x_0^1; x_0^2) - \mathbb{E}_{p(x_0^2)}\mathbb{E}_{p(x_t^2|x_0^2)}\left[KL(p(x_0^1|x_0^2)\|p(x_0^1|x_t^2))\right], \tag{5}$$

$$\mathcal{I}(x_t^1; x_t^2) \approx \mathcal{I}(x_0^1; x_0^2) - \mathbb{E}_{p(x_0^2)}\mathbb{E}_{p(x_t^2|x_0^2)}\left[KL(p(x_0^1|x_0^2)\|p(x_0^1|x_t^2))\right]$$
$$- \mathbb{E}_{p(x_0^1)}\mathbb{E}_{p(x_t^1|x_0^1)}\left[KL(p(x_t^2|x_0^1)\|p(x_t^2|x_t^1))\right]. \tag{6}$$

Proposition 2 suggests that the mutual information $\mathcal{I}(x_0^1; x_t^2)$ and $\mathcal{I}(x_t^1; x_t^2)$ are lower than the ideal mutual information $\mathcal{I}(x_0^1; x_0^2)$. With the growth of $t$, the KL divergence terms in (5) and (6) increase due to larger noise perturbation on $x_0^1$ and $x_0^2$. Thus, the gap between $\mathcal{I}(x_0^1; x_t^2)$, $\mathcal{I}(x_t^1; x_t^2)$ and $\mathcal{I}(x_0^1; x_0^2)$ becomes larger and the ability to extract contextual information is degraded.

**Mask graph.** We further analyze the mask-reconstruction via contrastive objectives in mask graphs. In U-MAE [46], the mask-reconstruction can be transformed into a constrastive learning objective and there exists a bipartite mask graph to elaborate this transformation [46]. The mask graph is consistent with $\mathcal{L}_{asym}$ in Proposition 1. Note that MaskDiT, SD-DiT, and MDT share the same mask graph, since their inputs are all unmasked noisy patches and noisy masked patches. Figure 1(b) illustrates the mask graph for clean image reconstruction in MAE (top) and that for noisy patch reconstruction in MaskDiT, SD-DiT, and MDT (bottom). In Proposition 3, we prove that contrastive objective between noisy patches could interfere learning contextual information.

**Proposition 3** *The asymmetric loss of noisy patch reconstruction and the asymmetric loss of clean patch reconstruction satisfy that*

$$\mathcal{L}_{asym-NN} = -\mathbb{E}_{p(x_t^1, x_t^2)}\left[h(x_t^1)^T h_g(x_t^2)\right]$$
$$\approx \mathcal{L}_{asym} + \mathbb{E}\left\{-h(x_t^1)^T \left[\frac{\partial h_g}{\partial x_0^2}\right]^T n\right\} + \mathbb{E}\left\{-h_g(x_0^2)^T \left[\frac{\partial h}{\partial x_0^1}\right]^T n\right\}, \tag{7}$$

*where $\mathcal{L}_{asym}$ is defined in* (1) *and represents contextual information. The two noise-weighted items represent contrastive objective between $h(x_t^1)$ and $[\partial h_g/\partial x_0^2]$ ($h_g(x_0^2)$ and $[\partial h/\partial x_0^1]$) weighted by the Gaussian noise $n$.*

Proposition 3 implies that the Gaussian noise introduces two extra terms in (7) and could affect the optimization process of $\mathcal{L}_{asym}$. Noisy patch reconstruction undermines the contrastive objective of contextual information, since larger Gaussian noise leads to more severe perturbation on $\mathcal{L}_{asym}$.

In summary, we leverage mutual information and contrastive asymmetric loss to demonstrate that the noisy patches mask-reconstruction is sub-optimal to learn real contextual information and larger noise could have more serious impact on context information extraction. This is consistent with the results in Figure 1(a). To solve this problem, in Section 3.3, we propose MC-DiT to effectively reconstruct masked clean patches from unmasked clean patches.

### 3.3 Contextual Enhancement with Masked Clean Patches

As demonstrated in Propositions 2 and 3 that **reconstructing masked noisy patches from unmasked noisy patches is insufficient for contextual information extraction**, we propose a novel method named MC-DiT to enhance contextual information extraction for DiT from the perspective of leveraging masked clean patches to reconstruct unmasked clean patches. Figure 2(a) depicts the proposed framework for MC-DiT. The clean images are disrupted by Gaussian noise $n \sim \mathcal{N}(0, t^2 I)$, where $t$ is the time step. Then the noisy images are patchified and masked by a random binary mask $m$. The unmasked noisy patches $x_t^1$ are fed into the DiT encoder for feature extraction. For contextual information extraction, the masked clean patches $x_0^2$ are concatenated with extracted feature $z = concat(z_1, x_0^2)$, where $z_1$ is the feature of $x_t^1$. After that, the feature $z$ is sent to DiT decoder to reconstruct clean unmasked patches $x_0^1$, which is consistent with previous masked diffusion ([48],[49]). The training objective of unmasked clean patch reconstruction is:

$$\mathcal{L}_{clean} = \mathbb{E}_{x_0 \sim p_{data}}\mathbb{E}_{n \sim \mathcal{N}(0, t^2 I)}\|(D_\theta((x_0 + n) \odot (1 - m), x_0 \odot m, t) - x_0) \odot (1 - m)\|^2. \tag{8}$$

By applying masked clean patches $x_0^2$ in (8), the information in $x_0^2$ is transferred to unmasked clean output patches $\tilde{x}^1$, which is constrained to equal $x_0^1$. It is not disrupted by noise $n$, since there is no

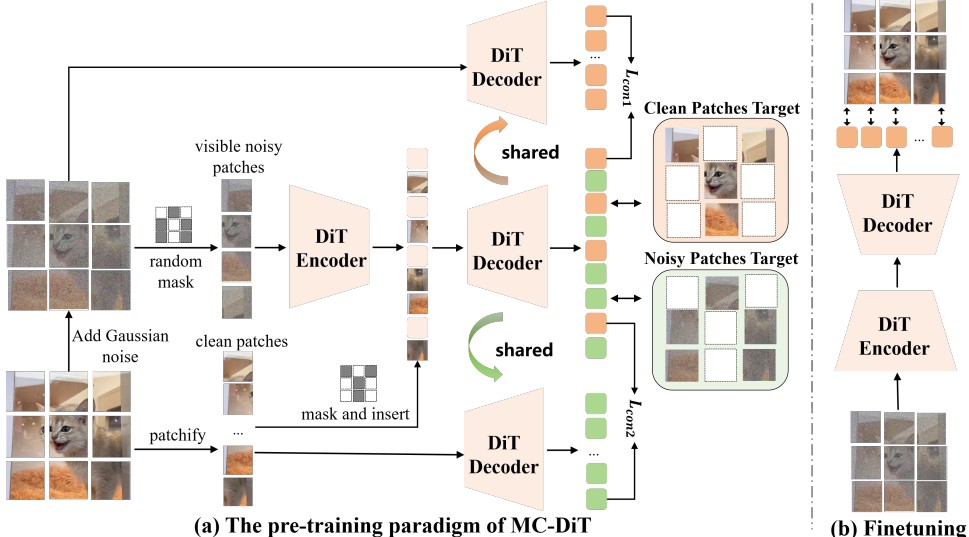

**(a) The pre-training paradigm of MC-DiT** **(b) Finetuning**

Figure 2: Framework of the proposed MC-DiT. (a) Pre-training. MC-DiT introduces unmasked clean patches and learns sufficient contextual information by reconstructing unmasked clean patches from masked clean patches. Two complementary EMA branches are developed to avoid model collapse. (b) Finetuning. MC-DiT is trained with unmasked patches for denoising.

noise in $x_0^2$ and $x_0^1$. Moreover, as mentioned in Section 3.2, the mutual information for our MC-DiT is $\mathcal{I}(x_0^1; x_0^2)$, since we leverage masked clean patches to recover unmasked clean patches. According to Proposition 2, $\mathcal{I}(x_0^1; x_0^2)$ is much higher than $\mathcal{I}(x_0^1; x_t^2)$ and $\mathcal{I}(x_t^1; x_t^2)$ under different noise $n$ and time steps $t$. Thus, the mutual information learned by our MC-DiT would not decrease for different noise at different time steps and sufficient contextual information could be learned for reconstruction.

In addition, we explore the potential benefits of $x_0^2$ by enforcing the masked output patches $\tilde{x}^2$ to match $x_t^2$. As discussed in Section 3.1, the denoising network $D_\theta$ predicts clean images from noisy images and can be viewed as distinguishing the clean images and noise. Therefore, predicting $x_t^2$ from $x_0^2$ is to recognize the noise $n$ and add to $x_0^2$. To this end, we employ reversed constraint on masked output patches $\tilde{x}^2$, as illustrated in Figure 2(a). The training objective is

$$\mathcal{L}_{noise} = \mathbb{E}_{x_0 \sim p_{data}} \mathbb{E}_{n \sim \mathcal{N}(0, t^2 I)} \|(D_\theta((x_0 + n) \odot (1 - m), x_0 \odot m, t) - (x_0 + n)) \odot m\|^2. \quad (9)$$

### 3.4 Addressing Model Collapse

Although MC-DiT can learn sufficient contextual information in theory, there exists model collapse problem in practice. The model learns a shortcut way that it only leverages masked clean patches to reconstruct unmasked clean patches and neglect the unmasked noisy patches. We further address the model collapse problem caused by only using masked clean patches to reconstruct unmasked clean patches for training. We introduce two extra EMA (Exponential Moving Average) branches of DiT decoders[3] to balance the mask-reconstruction and denoising objective. As shown in Figure 2(a), given the noisy input to DiT encoder, we introduce two branches to achieve only mask-reconstruction and denoising alongside the original DiT decoder. The upper branch of DiT decoder $D_\phi$ reconstructs from the whole noisy patches and captures denoising information, while the bottom branch processes clean patches and captures contextual information $D_\varphi$. Constraints on the two branches are employed in the loss function to balance the DiT decoder.

$$\mathcal{L}_{con1} = \mathbb{E}_{x_0 \sim p_{data}} \mathbb{E}_{n \sim \mathcal{N}(0, t^2 I)} \|(D_\theta((x_0 + n) \odot (1 - m), x_0 \odot m, t) - D_\phi(x_0 + n)) \odot (1 - m)\|^2, \quad (10)$$

$$\mathcal{L}_{con2} = \mathbb{E}_{x_0 \sim p_{data}} \mathbb{E}_{n \sim \mathcal{N}(0, t^2 I)} \|(D_\theta((x_0 + n) \odot (1 - m), x_0 \odot m, t) - D_\varphi(x_0)) \odot (1 - m)\|^2. \quad (11)$$

---

[3] The parameters of EMA decoders are initialized with those in the DiT decoders and are updated in the EMA fashion according to the parameters in DiT decoders: $\theta_{ema} = \alpha \times \theta_{ema} + (1 - \alpha) \times \theta_{dec}$, where $\alpha$ denotes the weight coefficient. The two decoders are updated only using the EMA method without using gradient updates.

Table 1: Comparison with state-of-the-art approaches for ImageNet-256×256 class conditional image generation. Bold font represents the best result. '-G' means using classifier-free guidance.

| Methods | FID ↓ | sFID ↓ | IS ↑ | Prec. ↑ | Rec. ↑ |
|---|---|---|---|---|---|
| BiGAN-deep [5] | 6.95 | 7.36 | 171.40 | 0.87 | 0.28 |
| StyleGAN-XL [41] | 2.30 | 4.02 | 265.12 | 0.78 | 0.53 |
| MaskGIT [6] | 6.18 | - | 182.10 | 0.80 | 0.51 |
| CDM [20] | 4.88 | - | 158.71 | - | - |
| ADM [9] | 10.94 | 6.02 | 100.98 | 0.69 | 0.63 |
| ADM-U [9] | 7.49 | **5.13** | 127.49 | 0.72 | 0.63 |
| LDM-8 [37] | 15.51 | - | 79.03 | 0.65 | 0.63 |
| LDM-4 [37] | 10.56 | - | 209.52 | **0.84** | 0.35 |
| U-ViT-H/2 [2] | 6.58 | - | - | - | - |
| DiT-XL/2 [34] | 9.62 | 6.85 | 121.50 | 0.67 | **0.67** |
| MDT-XL/2 [13] | 6.23 | 5.23 | 143.02 | 0.71 | 0.65 |
| MaskDiT-XL/2 [48] | 5.69 | 10.34 | 177.99 | 0.74 | 0.60 |
| SD-DiT-XL/2 [49] | 7.21 | 5.17 | 144.68 | 0.72 | 0.61 |
| MC-DiT-XL/2 | **4.14** | 6.96 | **309.69** | 0.83 | 0.62 |
| ADM-G [9] | 4.59 | 5.25 | 186.70 | 0.82 | 0.52 |
| ADM-U-G [9] | 3.94 | 6.14 | 215.84 | 0.83 | 0.53 |
| LDM-8-G[37] | 7.76 | - | 103.49 | 0.71 | 0.62 |
| LDM-4-G [37] | 3.60 | - | 247.67 | **0.87** | 0.48 |
| U-ViT-H/2-G[2] | 2.29 | 5.68 | 263.88 | 0.82 | 0.57 |
| DiT-XL/2-G [34] | 2.27 | 4.60 | 278.24 | 0.83 | 0.57 |
| MDT-XL/2-G [13] | 1.79 | **4.57** | 283.01 | 0.81 | 0.61 |
| MaskDiT-XL/2-G [48] | 2.28 | 5.67 | 276.56 | 0.80 | 0.61 |
| MC-DiT-XL/2-G | **1.78** | 4.87 | **290.17** | 0.81 | **0.62** |

Here, $D_\phi$ and $D_\varphi$ are two EMA DiT decoders. Therefore, the overall loss is:

$$\mathcal{L} = \mathcal{L}_{clean} + \lambda_1 \mathcal{L}_{noise} + \lambda_2 \mathcal{L}_{con1} + \lambda_3 \mathcal{L}_{con2}, \tag{12}$$

where $\lambda_1$, $\lambda_2$ and $\lambda_3$ are hyper-parameters.

**Unmasking Finetuning.** Similar to MaskDiT [48], although our MC-DiT captures contextual information during masking training, directly applying the pretrained model in inference leads to unsatisfactory performance, which is caused by the training-inference discrepancy. The clean patches used in training are not provided in the inference time. Thus, after training our MC-DiT, we finetune it on the unmasked scenarios for better performance, as shown in Figure 2 (b). It is worth noting that MC-DiT only needs a few iteration in the finetuning to generated semantic coherence images.

## 4 Experiments

### 4.1 Implementation Details

**Model Settings.** We follow the same architecture in MaskDiT [48]. Specifically, we first apply a pretrained variational autoencoder (VAE) from Stable Diffuion [37] to map the images into latent space, and then train our MC-DiT to reconstruct clean patches from noisy patches under the EDM [21] framework to approximate score function in the diffusion process. The pretrained VAE maps 256×256×3 input images to 32×32×4 latent features and 512×512×3 images to 64×64×4 latent features. Similar to SD-DiT, we apply DiT-S, DiT-B, and DiT-XL as our backbones.

**Training Details.** Similar to previous works [13, 48, 49], we train MC-DiT on ImageNet [39] with resolutions 256×256×3 and 512×512×3, respectively. Most training settings are the same as MaskDiT [48]. We train MC-DiT for 400K to 1M iterations using the AdamW optimizer with learning rate 0.0001 and no weight decay. By default, we use 50% mask ratio and batch size 1024. $\lambda_1$ and $\lambda_2$ in (12) are set to 0.1 and 0.05 for more denoising reconstruction. The EMA coefficient is set to 0.999 for smoothness and no data augmentation is employed.

**Evaluation Metrics.** Following DiT [34], we leverage Fréchet Inception Distance (FID) to measure the quality of generated images. For fair comparison, we also use ADM's Tensorflow evaluation

Table 2: Comparison with state-of-the-art approaches for ImageNet-512×512 class conditional image generation. The bold font represents the best performance.

| Methods | FID ↓ | sFID ↓ | IS ↑ | Prec. ↑ | Rec. ↑ |
|---|---|---|---|---|---|
| BiGAN-deep [5] | 8.43 | 8.13 | 177.90 | 0.88 | 0.29 |
| StyleGAN-XL [41] | 2.41 | 4.06 | 267.75 | 0.77 | 0.52 |
| ADM [9] | 23.24 | 10.19 | 58.06 | 0.73 | 0.60 |
| ADM-U [9] | 9.96 | **5.62** | 121.78 | 0.75 | **0.64** |
| DiT-XL/2 [34] | 12.03 | 7.12 | 105.25 | 0.75 | **0.64** |
| MaskDiT-XL/2 [48] | 10.79 | 13.41 | 145.08 | 0.74 | 0.56 |
| MC-DiT-XL/2 | **9.30** | 6.28 | **179.58** | **0.76** | 0.53 |
| ADM-G[9] | 7.72 | 6.57 | 172.71 | **0.87** | 0.42 |
| ADM-U-G[9] | 3.85 | 5.86 | 221.72 | 0.84 | 0.53 |
| DiT-XL/2-G [34] | 3.04 | 5.02 | 240.82 | 0.84 | 0.54 |
| MaskDiT-XL/2-G [48] | 2.50 | 5.10 | 256.27 | 0.83 | 0.56 |
| MC-DiT-XL/2-G | **2.03** | **4.87** | **272.19** | 0.84 | **0.56** |

Table 3: Comparison with state-of-the-art approaches ImageNet-256×256 class conditional image generation at different scales and iterations. '-S', '-B', '-XL' means 'small', 'base' and largest model size, respectively and '/2' denotes the patch size of 2 for all input patches.

| Methods | Training Iterations | FID-50K ↓ |
|---|---|---|
| DiT-S/2 [34] | 400K | 68.40 |
| MDT-S/2 [13] | 400K | 53.46 |
| SD-DiT-S/2 [49] | 400K | 48.39 |
| MC-DiT-S/2 | 400K | 41.67 |
| DiT-B/2 [34] | 400K | 43.47 |
| MDT-B/2 [13] | 400K | 34.33 |
| SD-DiT-B/2 [49] | 400K | 28.62 |
| MC-DiT-B/2 | 400K | 18.88 |
| DiT-XL/2 [34] | 7000K | 9.62 |
| MaskDiT-XL/2 [48] | 1300K | 12.15 |
| MDT-XL/2 [13] | 1300K | 9.60 |
| SD-DiT-XL/2 [49] | 1300K | 9.01 |
| MC-DiT-XL/2 | 1300K | 7.92 |

suite [9] to compute FID-50K (FID for short), sFID [31], Inception Score (IS) [40] and Precision/Recall [24] as secondary metrics. More vivid images have lower FID and sFID, while their IS and Precision/Recall are higher.

## 4.2 Experimental Results

We evaluate vanilla training (*i.e.*, LDM [37], ADM [9], and DiT [34]) and masked training (*i.e.*, proposed MC-Dit, MaskDiT[48], MDT [13], and SD-DiT [49]) using backbones of different scales for 256×256 and 512×512 image generation on ImageNet.

**Results on ImageNet-256×256.** Table 1 shows that our MC-DiT-XL/2 achieves the smallest FID score and the highest IS score. Compared with non-masked diffusion models, MC-DiT-XL/2 decrease the FID score from 9.62 to 4.14. Compared with masked diffusion models, the FID score decreases from 5.69 to 4.14. With classifier-free guidance (-G), our MC-DiT-XL/2-G achieves the best FID score of 1.78, and the highest IS score, which significantly outperforms previous works.

**Results on ImageNet-512×512.** Table 2 shows that MC-DiT-XL/2 achieves a FID of 9.30 and outperforms MaskDiT [48] and DiT [34]. The IS score of MC-DiT is also the highest, indicating the effectiveness of our method. With classifier-free guidance (-G), our MC-DiT-XL/2-G achieves the best FID score of 2.03, indicating the effectiveness of MC-DiT.

**Contextual Enhancement.** Figure 1 (a) reports the mutual information between unmasked and masked output patches with different noise, which can be viewed as the metric of contextual information consistency. Our MC-DiT decreases slowly during the noise variance becomes larger, which indicates more sufficient contextual reconstruction regardless noise.

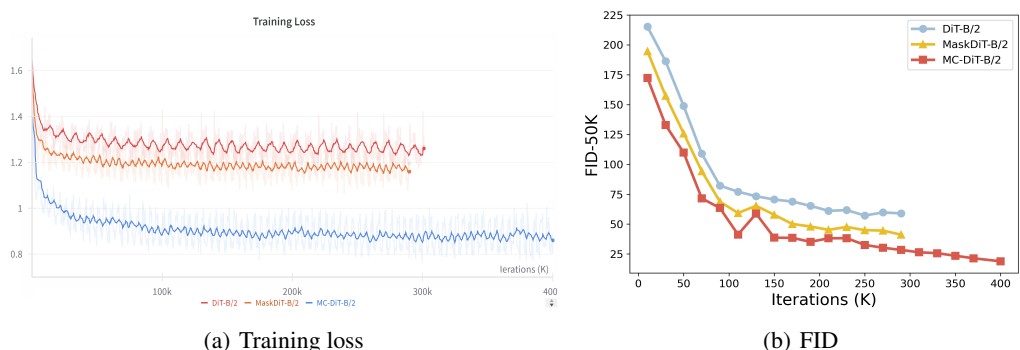

| (a) Training loss | (b) FID |
|---|---|

Figure 3: Training loss and FID for DiT-B/2, MaskDit-B/2, and MC-DiT-B/2 during training.

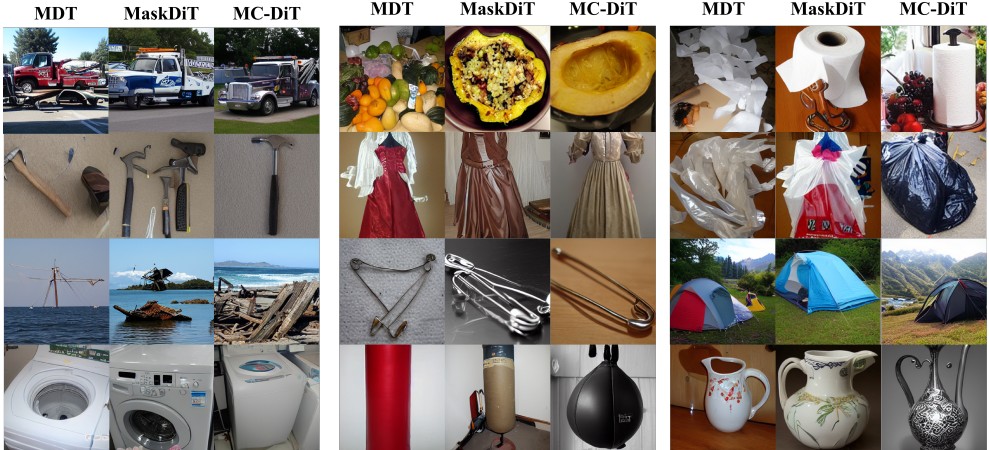

Figure 4: Comparison of $256 \times 256$ images generated by MDT, MaskDiT and MC-DiT. Various details are strange in images generated by MDT and MaskDiT.

**Backbones at different scales.** Table 3 evaluates FID-50K at different scales and training iterations with various backbones. Notice that MaskDiT only reports the performance of 'XL' scale. Under fixed number of training iterations, MC-DiT outperforms vanilla DiT [34], MDT [13], and SD-DiT [49] in FID by a large margin, *i.e.*, 6.72, 9.74, and 1.09 FID reduction for DiT-S, DiT-B, and DiT-XL backbones. This fair comparison fully demonstrates the effectiveness of our method.

**Convergence speed.** In order to evaluate the convergence speed of various methods, we compare the training loss curve in Figure 3(a). We report the MSE loss (Eq. (8)) on clean patches for fairness. We train MaskDiT [48] and DiT [34] for 300K iterations due to the substantial time and GPU resource overhead and use the training curve of our MC-DiT trained for evaluations, which is trained for 400K iterations. The training loss of MC-DiT decreases faster than DiT [34] and MaskDiT [48]. Figure 3(b) measures FID-50K at each step after unmasked tuning and shows that MC-DiT achieves the lowest FID-50K score.

**Generated image comparison.** Figure 4 visualizes the $256 \times 256$ images generated by MDT [13], MaskDiT[48] and our MC-DiT. Our generated images are more realistic and have more consistent textual structure than MaskDiT and MDT. For example, images of 'hammer' generated by MaskDiT and MDT have incomplete structure, while our MC-DiT generates images with more complete structures, validating the superior contextual information extraction ability of our MC-DiT.

### 4.3 Ablation Studies

For computation efficiency, we adopt '-B' in all the models for fair comparison. All the models are trained for 400K itreations with batch size 256 and mask ratio $50\%$.

Table 4: Ablation study of hyperparameters.

| $\lambda_1$ | FID | $\lambda_2$ | FID | $\lambda_3$ | FID |
|---|---|---|---|---|---|
| 0 | 43.23 | 0.0 | 38.83 | 0.0 | 37.77 |
| 0.01 | 40.99 | 0.01 | 36.15 | 0.05 | 35.20 |
| 0.1 | 35.20 | 0.1 | 35.20 | 0.1 | 35.46 |
| 1.0 | 38.97 | 1.0 | 37.54 | 1.0 | 36.35 |

Table 5: Ablation study of the EMA branches.

| Branches | FID |
|---|---|
| Main Branch | 22.10 |
| w Noisy Branch | 19.26 |
| w Clean Branch | 18.88 |

Table 6: Comparison with different targets.

| Reconstruction Target | FID-50K ↓ |
|---|---|
| All the clean patches | 34.53 |
| Only clean patches | 25.88 |
| Clean patches + Noisy patches | 22.10 |

Table 7: Ablation study of unmasking tuning.

| Strategy | Iterations | FID |
|---|---|---|
| MaskDiT-XL/2 w UT | 1300K | 12.15 |
| MC-DiT-XL/2 w UT | 1300K | 7.92 |
| MC-DiT-XL/2 w/o UT | 1300K | 8.33 |

**Main branch target.** We evaluate the reconstruction targets of main branch by considering three cases, *i.e.*, clean patch reconstruction + noisy patch reconstruction, all clean patch reconstruction and only clean patch reconstruction. 'All clean patches' means all the patches are constrained by clean reconstruction loss. 'Only clean patches' means only unmasked patches are constrained by clean reconstruction loss, and 'Clean patches + Noisy patches' means masked patches are further constrained by noisy reconstruction loss. Table 6 shows that our model performs the best, which validates the effectiveness of noisy reconstruction loss.

**Effectiveness of two extra EMA branches.** Table 5 evaluates the influence of two EMA branches. FID decreases obviously by 2.84 using the noise branch, indicating the necessity of noisy branch to address model collapse. Further experiments can be found in appendix.

**Unmasked tuning.** Unmasked tuning (UT) can reduce the training-inference discrepancy, as demonstrated by MaskDiT and is adopted in our MC-DiT. However, we can remove unmasked tuning to reduce the complexity at little loss on FID. Table 7 shows that FID will increase by only 0.41 for MC-DiT by removing unmasked tuning.

**Hyperparameters.** We separately evaluate four values for $\lambda_1$, $\lambda_2$, and $\lambda_3$. Table 4 shows best FID is obtained when $\lambda_1 = 0.1$, $\lambda_2 = 0.1$, and $\lambda_3 = 0.05$. Note that $\lambda_2$ is larger than $\lambda_3$ since the denoising objective is more important than contextual information utilization.

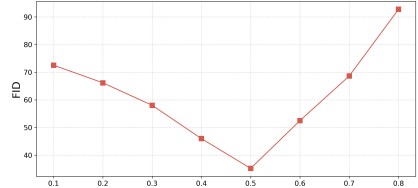

Figure 5: Ablation study of mask ratio.

**Mask ratio.** Figure 5 visualizes the influence of the mask ratio in $m$. FID is smallest at the mask ratio of 50% and increases rapidly when the mask ratio is larger than 50%.

## 5 Conclusion

In this paper, we summarize the previous works that combine mask-reconstruction with DiT training and claim that reconstructing masked noisy patches from unmasked noisy patches is insufficient for contextual information extraction. To validate this claim, we analyze the mutual information and contrastive objective theoretically and experimentally. Besides, we propose a new pretraining paradigm (dubbed MD-DiT), which reconstructs unmasked clean patches from masked clean patches and guarantees the contextual information extraction. Moreover, to avoid model collapse, two extra EMA branches are applied in MC-DiT to adjust the balance between the mask-reconstruction task and denoising objective. Extensive experiments demonstrate the robustness of our method and our MC-DiT achieves the state-of-the-art performance in image generation.

**Limitations.** Despite excellent performance, the training speed and inference speed of MC-DiT still needs to be improved. We will mitigate this issue in future work by transferring the information in the encoder into the decoder, which decreases the training difficulty.

**Acknowledgment.** This work was supported in part by the National Natural Science Foundation of China under Grant 62125109, Grant T2122024, Grant 62320106003, Grant 62371288, Grant 62431017, Grant 62401357, Grant 62401366, Grant 61931023, Grant 61932022, Grant 62120106007.

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

# A  Supplemental Material

## A.1  Theoretical Proof

**Proposition 2.** *Given masked and unmasked clean patches $x_0^1$ and $x_0^2$ and their noisy versions $x_t^1$ and $x_t^2$, the mutual information $\mathcal{I}(x_t^1; x_t^2)$, $\mathcal{I}(x_0^1; x_t^2)$, and $\mathcal{I}(x_0^1; x_0^2)$ satisfy that*

$$\mathcal{I}(x_0^1; x_t^2) \approx \mathcal{I}(x_0^1; x_0^2) - \mathbb{E}_{p(x_0^2)}\mathbb{E}_{p(x_t^2|x_0^2)}\left[KL(p(x_0^1|x_0^2)\|p(x_0^1|x_t^2))\right] \tag{13}$$

$$\mathcal{I}(x_t^1; x_t^2) \approx \mathcal{I}(x_0^1; x_0^2) - \mathbb{E}_{p(x_0^2)}\mathbb{E}_{p(x_t^2|x_0^2)}\left[KL(p(x_0^1|x_0^2)\|p(x_0^1|x_t^2))\right]$$

$$- \mathbb{E}_{p(x_0^1)}\mathbb{E}_{p(x_t^1|x_0^1)}\left[KL(p(x_t^2|x_0^1)\|p(x_t^2|x_t^1))\right] \tag{14}$$

**Proof.** Given the time step $t$, masked noisy patches $x_t^2$, masked clean patches $x_0^2$, clean unmasked patches $x_0^1$ and noisy unmasked patches $x_t^1$, where $x_t^2 = x_0^2 + n, x_t^1 = x_0^1 + n$ and $n \sim \mathcal{N}(0, t^2 I)$. we derive the mutual information $\mathcal{I}(x_0^1; x_t^2)$ according to the definition.

$$\mathcal{I}(x_0^1; x_t^2) = \mathbb{E}_{p(x_0^1)}\mathbb{E}_{p(x_t^2|x_0^1)} \log \frac{p(x_0^1|x_t^2)}{p(x_0^1)} \tag{15}$$

$$= \mathbb{E}_{p(x_0^1)}\mathbb{E}_{p(x_t^2|x_0^1)} \log \left( \frac{p(x_0^1|x_0^2)}{p(x_0^1)} \cdot \frac{p(x_0^1|x_t^2)}{p(x_0^1|x_0^2)} \right) \tag{16}$$

$$\approx \mathbb{E}_{p(x_0^1)}\mathbb{E}_{p(x_0^2|x_0^1)}\mathbb{E}_{p(x_t^2|x_0^2)} \log \left( \frac{p(x_0^1|x_0^2)}{p(x_0^1)} \cdot \frac{p(x_0^1|x_t^2)}{p(x_0^1|x_0^2)} \right) \tag{17}$$

$$= \mathbb{E}_{p(x_0^1)}\mathbb{E}_{p(x_0^2|x_0^1)} \log \frac{p(x_0^1|x_0^2)}{p(x_0^1)} + \mathbb{E}_{p(x_0^1,x_0^2,x_t^2)} \log \frac{p(x_0^1|x_t^2)}{p(x_0^1|x_0^2)} \tag{18}$$

$$= \mathcal{I}(x_0^1; x_0^2) + \mathbb{E}_{p(x_0^2)}\mathbb{E}_{p(x_t^2|x_0^2)}\mathbb{E}_{p(x_0^1|x_0^2)} \log \frac{p(x_0^1|x_t^2)}{p(x_0^1|x_0^2)} \tag{19}$$

$$= \mathcal{I}(x_0^1; x_0^2) - \mathbb{E}_{p(x_0^2)}\mathbb{E}_{p(x_t^2|x_0^2)}\left[KL(p(x_0^1|x_0^2)\|p(x_0^1|x_t^2))\right] \tag{20}$$

Thus, the mutual information between noisy masked patches and unmasked clean patches $\mathcal{I}(x_0^1; x_t^2)$ is less than $\mathcal{I}(x_0^1; x_0^2)$ due to the non-negativity of KL divergence. Moreover, during $t$ increases, the variance of the Gaussian noise $n$ becomes larger. As a result, noisy masked patches $x_t^2$ are disrupted heavily from clean patches $x_0^2$. The distribution $p(x_0^1|x_t^2)$ is very dissimilar from $p(x_0^1|x_0^2)$. Formally, the derivation of KL divergency can be written as:

$$\mathbb{E}_{p(x_0^2)}\mathbb{E}_{p(x_t^2|x_0^2)}\mathbb{E}_{p(x_0^1|x_0^2)} \log \left[ \frac{p(x_0^1|x_t^2)}{p(x_0^1|x_0^2)} \right]$$

$$= \mathbb{E}_{p(x_0^2)}\mathbb{E}_{p(x_t^2|x_0^2)}\mathbb{E}_{p(x_0^1|x_0^2)} \log \left[ \frac{p(x_t^2|x_0^1)}{p(x_0^2|x_0^1)} \times \frac{p(x_t^2)}{p(x_0^2)} \right]$$

$$\approx \mathbb{E}_{p(x_0^2)}\mathbb{E}_{p(x_t^2|x_0^2)}\mathbb{E}_{p(x_0^1|x_0^2)} \log \left[ \frac{p(x_0^2|x_0^1) + p(n|x_0^1)}{p(x_0^2|x_0^1)} \times \frac{p(x_0^2) + p(n)}{p(x_0^2)} \right]$$

$$= \mathbb{E}_{p(x_0^2)}\mathbb{E}_{p(x_t^2|x_0^2)}\mathbb{E}_{p(x_0^1|x_0^2)} \log \left[ \left( 1 + \frac{p(n|x_0^1)}{p(x_0^2|x_0^1)} \right) \times \left( 1 + \frac{p(n)}{p(x_0^2)} \right) \right], \tag{21}$$

We approximate $p(x_t^2) \approx p(x_0^2) + p(n)$, since $p(x_t^2)$ is a Gaussion distribution with mean value $x_0^2$ and variance $t^2$. As $t$ increases, the KL divergence $KL(p(x_0^1|x_0^2)\|p(x_0^1|x_t^2))$ increases and the mutual information $\mathcal{I}(x_0^1; x_t^2)$ achieves the larger difference with $\mathcal{I}(x_0^1; x_0^2)$. Thus, the mutual information $\mathcal{I}(x_0^1; x_t^2)$ is lower than $\mathcal{I}(x_0^1; x_0^2)$.

Similarly, the mutual information between noisy patches $\mathcal{I}(x_t^1; x_t^2)$ can be drived according to Eq. 20:

$$\mathcal{I}(x_t^1; x_t^2) \approx \mathcal{I}(x_0^1; x_t^2) - \mathbb{E}_{p(x_0^1)}\mathbb{E}_{p(x_t^1|x_0^1)}\left[KL(p(x_t^2|x_0^1)\|p(x_t^2|x_t^1))\right] \tag{22}$$

$$\approx \mathcal{I}(x_0^1; x_0^2) - \mathbb{E}_{p(x_0^2)}\mathbb{E}_{p(x_t^2|x_0^2)}\left[KL(p(x_0^1|x_0^2)\|p(x_0^1|x_t^2))\right]$$

$$- \mathbb{E}_{p(x_0^1)}\mathbb{E}_{p(x_t^1|x_0^1)}\left[KL(p(x_t^2|x_0^1)\|p(x_t^2|x_t^1))\right] \tag{23}$$

Therefore, the proposition has been proved.

**Proposition 3.** *The asymmetric loss of noisy patch reconstruction and the asymmetric loss of clean patch reconstruction satisfy that:*

$$\mathcal{L}_{asym-NN} = -\mathbb{E}_{p(x_t^1,x_t^2)}\left[h(x_t^1)^T h_g(x_t^2)\right]$$

$$\approx \mathcal{L}_{asym} + \mathbb{E}\left\{-h(x_t^1)^T\left[\frac{\partial h_g}{\partial x_0^2}\right]^T n\right\} + \mathbb{E}\left\{-h_g(x_0^2)^T\left[\frac{\partial h}{\partial x_0^1}\right]^T n\right\}, \qquad (24)$$

*where $\mathcal{L}_{asym}$ is defined in* (1) *and represents contextual information. The two noise-weighted items represent contrastive objective between $h(x_t^1) - [\partial h_g/\partial x_0^2]$ and $h_g(x_0^2) - [\partial h/\partial x_0^1]$ weighted by the Gaussian noise $n$.*

**Proof.** According to Eq.1, the asymmetric loss can be written as:

$$\mathcal{L}_{asym-NN} = -\mathbb{E}_{p(x_t^1,x_t^2)}[h(x_t^1)^T h_g(x_t^2)] \qquad (25)$$

$$\approx -\mathbb{E}\left\{h(x_t^1)^T\left[h_g(x_0^2) + \left[\frac{\partial h_g}{\partial x_0^2}\right]^T n + o(x_0^2)\right]\right\} \qquad (26)$$

$$= -\mathbb{E}\left\{h(x_t^1)^T h_g(x_0^2) + h(x_t^1)^T\left[\frac{\partial h_g}{\partial x_0^2}\right]^T n\right\} \qquad (27)$$

$$\approx -\mathbb{E}\left\{\left[h(x_0^1) + \left[\frac{\partial h}{\partial x_0^1}\right]^T n + o(x_0^1)\right]^T h_g(x_0^2) + h(x_t^1)^T\left[\frac{\partial h_g}{\partial x_0^2}\right]^T n\right\} \qquad (28)$$

$$= -\mathbb{E}\left[h(x_0^1)^T h_g(x_0^2)\right] + \mathbb{E}\left\{-h(x_t^1)^T\left[\frac{\partial h_g}{\partial x_0^2}\right]^T n\right\} + \mathbb{E}\left\{-h_g(x_0^2)^T\left[\frac{\partial h}{\partial x_0^1}\right]^T n\right\} \qquad (29)$$

$$= \mathcal{L}_{asym} + \mathbb{E}\left\{-h(x_t^1)^T\left[\frac{\partial h_g}{\partial x_0^2}\right]^T n\right\} + \mathbb{E}\left\{-h_g(x_0^2)^T\left[\frac{\partial h}{\partial x_0^1}\right]^T n\right\} \qquad (30)$$

where we leverage first-order Taylor's formula in Eq. 26 and Eq. 28 to calculate $h(x_t^1)$ and $h_g(x_t^2)$ at $x_0^1$ and $x_0^2$, since $x_t^1 = x_0^1 + n$ and $x_t^2 = x_0^2 + n$. $o$ denotes the the higher order infinitesimal.

## A.2 Experiment Details

**Diffusion Settings.** We leverage EDM [21] as our diffusion training framework, which predicts clean image patches from noisy images. For fair comparison, we use the default parameters in EDM (see [21] for more details). During inference, we generate conditional images from Gaussian noise via EDM-sampler [21]. Specifically, the time steps in the reverse process are set via $t_i = (t_{max}^{\frac{1}{\rho}} + \frac{i}{N-1}(t_{min}^{\frac{1}{\rho}} - t_{max}^{\frac{1}{\rho}}))^\rho$, where $N = 40$, $\rho = 7$, $t_{max} = 80$ and $t_{min} = 0.002$. Besides, the second-order correction is applied and the generated images are the average of first-order and second-order results.

**Training Details.** We follow the LDM [37] and adopt a pretrained VAE to firstly map the images into the latent spaces. The weight of pretrained VAE is from Stable Diffusion [37]. Then, we train the denoising models with these latent features. We leverage AdamW optimizer with learning rate 0.0001, batch size 256, and $50\%$ mask ratio. As for unmasking fintuning, we slightly change some hyper-parameters with learning rate 0.00005, batch size 128, mask ratio $0\%$. Some details can be found in Table 8.

## A.3 Supplementary Experiments

**Generalization Experiments.** We adopt the ImageNet dataset in the experiments for a fair comparison, since MaskDiT[48], SD-DiT[49] and MDT[13] are all evaluated on the ImageNet dataset[39]. In fact, our MC-DiT can be generalized to different domains or datasets for improved image generation

Table 8: Experimental details about MC-DiT.

| | MC-DiT-B/2 | MC-DiT-XL/2 | MC-DiT-XL/2 |
|---|---|---|---|
| Resolution | $256 \times 256$ | $256 \times 256$ | $512 \times 512$ |
| Training Time | 50h | 586h | 623h |
| Inference Time (50K images) | 12h | 8h | 15.2h |
| GPUs | $2\times$ RTX-3090 GPUs | $4\times$ V100 GPUs | $4\times$ V100 GPUs |
| Batch Size | $256 \times 2$ | $256 \times 4$ | $128 \times 4$ |
| Memory Usage per GPU | 17GB | 20GB | 27GB |

Table 9: Performance comparison on Cifar10 and CelebA dataset of MaskDiT and MC-DiT

| Cifar10 | | CelebA | |
|---|---|---|---|
| Methods | FID | Methods | FID |
| MaskDiT-B/2 | 11.52 | MaskDiT-B/2 | 7.14 |
| MC-DiT-B/2 | 9.28 | MC-DiT-B/2 | 5.36 |

Table 10: Performance comparisons with different branches.

| Branches | FID |
|---|---|
| Main Branch | 22.10 |
| w/ Noisy Branch | 19.26 |
| w/ Clean Branch | 18.88 |
| Main Branch (unmasked noisy patch only) | 25.72 |
| Main Branch (masked clean patch only) | 23.69 |
| Main Branch (unmasked noisy patch only) w/ Noisy Branch | 19.84 |
| Main Branch (unmasked noisy patch only) w/ Clean Branch | 19.57 |

due to the fact that it can extract contextual information from arbitrary images. Table 9 compares the performance of MaskDiT[48] and MC-DiT on the CIFAR-10 [23] and CelebA [26] (that collected for face anti-spoofing) datasets. Due to time limit, we train both MaskDiT[48] and MC-DiT for 200K iterations. Experimental results show that MC-DiT outperforms MaskDiT[48] on both datasets.

**Convergence Speed.** In Figure 3, we compare the training curve of DiT [34], MaskDiT [48] and MC-DiT and point out that the training loss of MC-DiT decreases faster than DiT [34] and MaskDiT [48]. This is due to the primary focus of our analysis is the overall effectiveness of the model. The blue line can achieve a lower loss, despite similar iterqation counts for flattening, highlighting the model's efficiency in reaching a more optimal solution. Besides, the loss reported in Figure 3(a) denotes the MSE loss $\mathcal{L}_{clean}$. Thus, lower MSE loss means the generated clean patches are more similar to the ground-truth. Moreover, MC-DiT achieves lower MSE loss with the same iterations with DiT [34] and MaskDiT [48], indicating the performant model with higher convergence speed.

**Main Branch Target.** The modal collapse occurs when the main branch only considers clean-to-clean mask-reconstruction for masked clean patches but ignores the denoising of unmasked noisy patches. We propose two EMA branches to balance the two tasks for the main branch. We use the noisy EMA branch to realize noisy-to-clean mapping for denoising, and the clean EMA branch to realize clean-to-clean mapping for mask-reconstruction (mask ratio is 0%). The two EMA branches constrain the output of the main branch (minimize the MSE loss between the outputs of the main branches and EMA branches) via three hyper-parameters, which leads to the balance on the denoising task and clean-to clean mask-reconstruction task.

To verify this, we report in Table 10 the FID score of the main branch with noisy and clean patch inputted only. The result of the main branch with unmasked noisy patches only is higher than that of masked clean patches, indicating the modal collapse problem. With noisy and clean branches, the FID score of the main branches decline distinctly, validating the effectiveness of the EMA branches.

**Ablation Study of hyperparameters.** Following MaskDiT[48], we select 0.01, 0.1, and 1.0 as the scaling values of three hyperparameters and supplement various values for ablation study. Table 13, Table 14 and Table 15 evaluate various values of the three hyperparameters and we find that the best FID is still obtained when $\lambda_1 = 0.1$, $\lambda_2 = 0.1$, and $\lambda_3 = 0.05$.

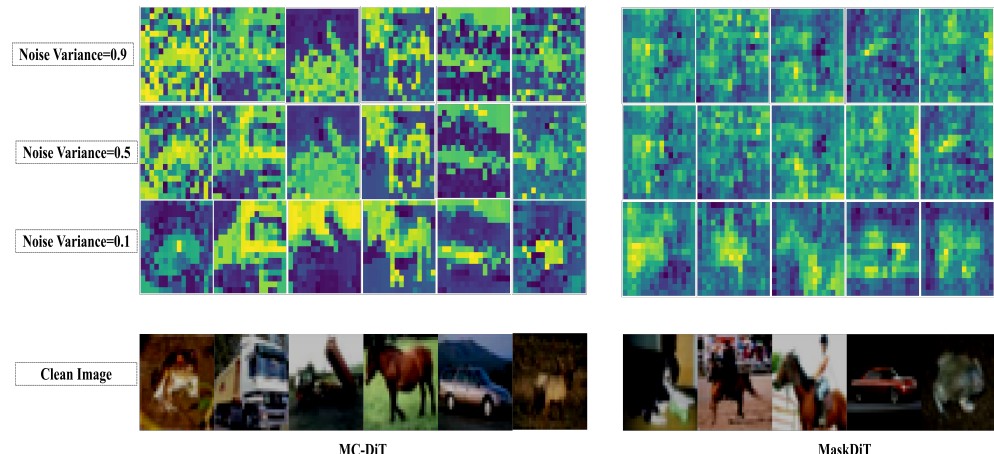

Figure 6: Feature visualization of MaskDiT and MC-DiT at different noise variance. Better viewed by zoom in.

Table 11: Parameters and training cost comparison between MDT, MaskDiT and MC-DiT. Training speed denotes the number of iterations per second.

| Setting | | | | | |
|---|---|---|---|---|---|
| $256 \times 256$ | | | | | |
| Modal | Params | FLOPs | Mem | Speed | FID |
| MDT-XL/2 | 742M | 28G | 20G | 1.22 | 6.23 |
| MaskDiT-XL/2 | 730M | 24G | 18G | 3.09 | 5.69 |
| MC-DiT-XL/2 | 786M | 26G | 20G | 1.45 | 4.14 |
| $512 \times 512$ | | | | | |
| Modal | Params | FLOPs | Mem | Speed | FID |
| MDT-XL/2 | 742M | 64G | 28G | 0.83 | - |
| MaskDiT-XL/2 | 730M | 56G | 24G | 1.98 | 10.79 |
| MC-DiT-XL/2 | 786M | 60G | 27G | 1.05 | 9.30 |

**Attention Map Visualization.** Figure 6 visualizes the attention map of MaskDiT and MC-DiT at different noise variance with Cifar10 dataset[23]. A larger noise variance denotes the noise with large scale. Our MC-DiT extracts proper shape for various noise scale, while the features extracted by MaskDiT are messy in the large noise scale. This further proves the motivation and effectiveness of our paper that clean-to-clean mask reconstruction promotes learning sufficient contextual information.

**Training cost comparison.** We compare the training cost (parameters, FLOPs, memory used and training speed) on $4\times$V100 GPUs in Table 11. The training speed of MC-DiT is a little bit slower than other methods due to two EMA branches. However, the inference speed of MC-DiT is similar to MaskDiT, since two EMA branches are removed during inference. The additional overhead of MC-DiT is relatively small ($7.6\%$ parameters and $8\%$ FLOPs), but the FID performance improvement is significant.

**EMA Branch with DiT encoder.** In the main branch of MC-DiT, the unmasked noisy patches are fed into the DiT encoder, while all the noisy patches are directly inserted into the EMA DiT decoder to avoid modal collapse, as shown in the Figure 2. The reasons are on the two folds: (1) efficient. Only apply DiT decoder for EMA branches leads to small extra parameters and fast inference speed, while EMA DiT encoder slows down the entire EMA branches. (2) effective. The DiT decoder is trained to extract masked clean images patches in the main branch. Thus, directly apply image patches as the input of EMA DiT decoder does not lead to poor denoising results. As shown in Table 12, applying EMA DiT encoder introduces extra 669M parameters, while FID score only decreases 1.35. Thus, to balance the parameters and performance, we select DiT decoder in the EMA branches.

Table 12: Performance and parameters comparisons with and without DiT encoder in EMA branches.

| Branch | Params | FID |
|---|---|---|
| DiT Decoder | 56M | 18.88 |
| DiT Decoder + DiT encoder | 725M | 17.53 |

Table 13: Ablation study on $\lambda_1$

| $\lambda_1$ | 0 | 0.01 | 0.03 | 0.05 | 0.07 | 0.09 | 0.1 | 0.3 | 0.5 | 0.7 | 0.9 | 1.0 |
|---|---|---|---|---|---|---|---|---|---|---|---|---|
| FID | 43.23 | 40.99 | 39.23 | 38.44 | 37.95 | 36.53 | 35.20 | 35.98 | 36.74 | 36.91 | 37.52 | 38.97 |

Table 14: Ablation study on $\lambda_2$

| $\lambda_2$ | 0 | 0.01 | 0.03 | 0.05 | 0.07 | 0.09 | 0.1 | 0.3 | 0.5 | 0.7 | 0.9 | 1.0 |
|---|---|---|---|---|---|---|---|---|---|---|---|---|
| FID | 38.83 | 36.15 | 36.02 | 36.46 | 35.99 | 36.07 | 35.20 | 35.34 | 36.18 | 37.26 | 35.98 | 37.54 |

Table 15: Ablation study on $\lambda_3$

| $\lambda_3$ | 0 | 0.01 | 0.03 | 0.05 | 0.07 | 0.09 | 0.1 | 0.3 | 0.5 | 0.7 | 0.9 | 1.0 |
|---|---|---|---|---|---|---|---|---|---|---|---|---|
| FID | 37.77 | 37.25 | 36.63 | 35.20 | 36.07 | 37.93 | 35.46 | 35.88 | 37.26 | 36.19 | 37.40 | 36.35 |

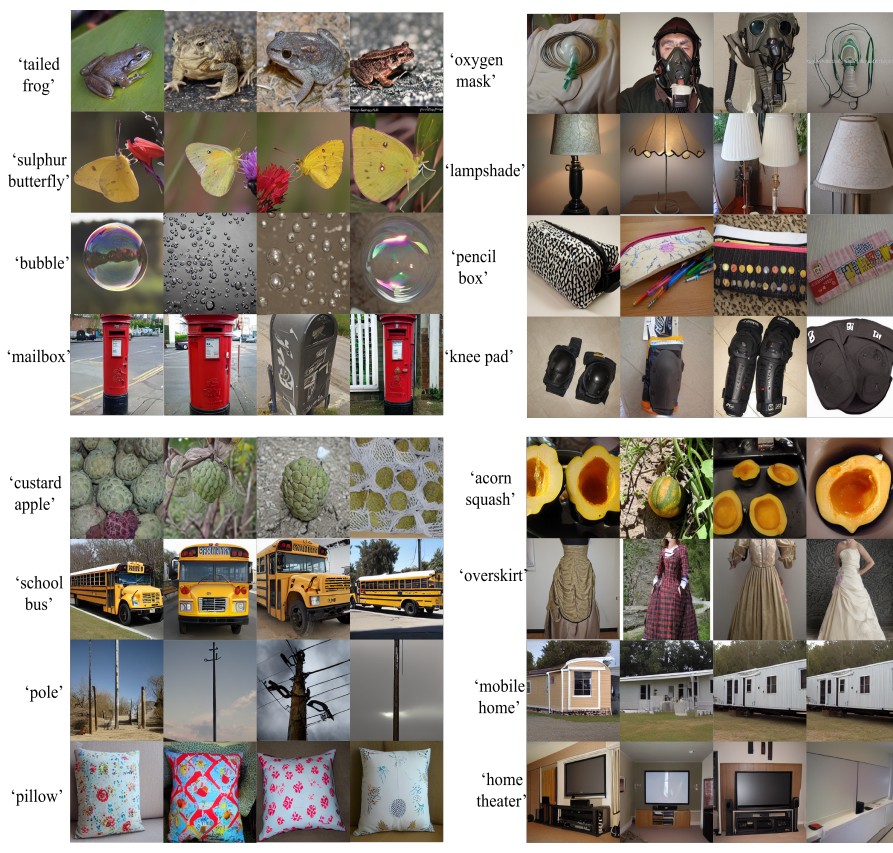

Figure 7: Visualization of $256 \times 256$ images generated by our MC-DiT.

## A.4  Generated Samples

Figure 7 visualizes some images generated by our MC-DiT with $256 \times 256$ resolutions. Figure 8 visualizes some images generated by our MC-DiT with $512 \times 512$ resolutions.

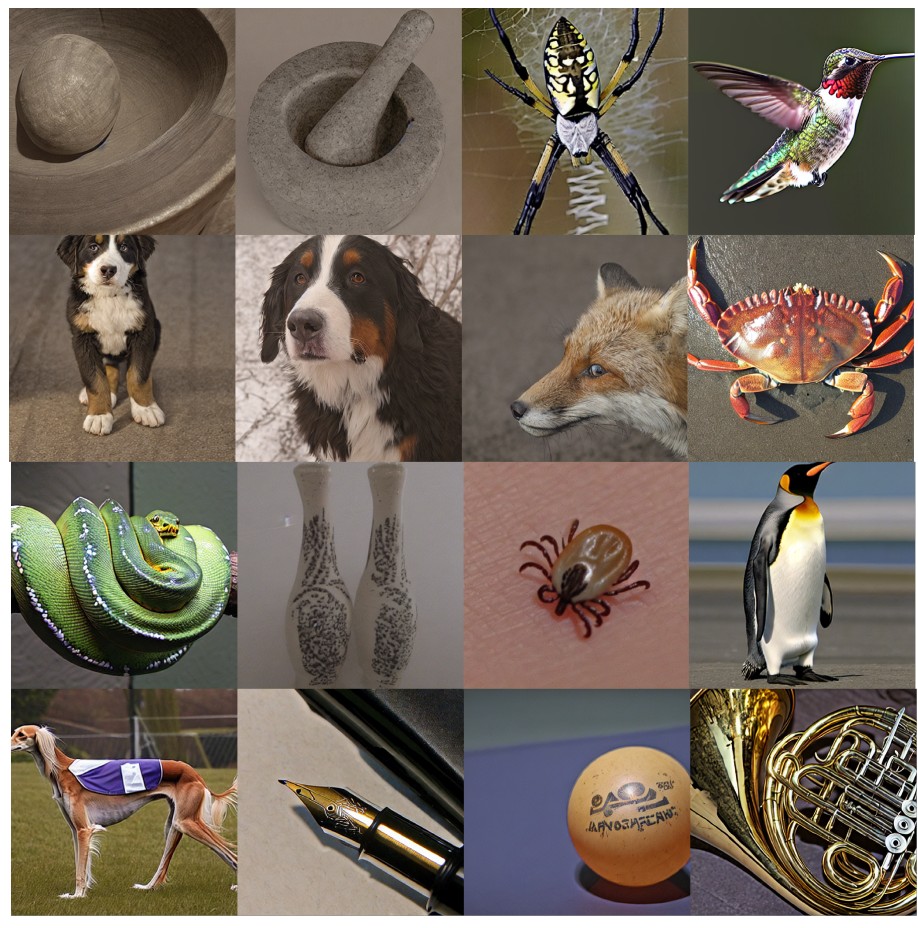

Figure 8: Visualization of $512 \times 512$ images generated by our MC-DiT

