# OpenReview forum: "MC-DiT: Contextual Enhancement via Clean-to-Clean Reconstruction for Masked Diffusion Models"
_NeurIPS.cc/2024/Conference — NeurIPS 2024 poster_

### Official Review · Reviewer_gqjf · 2024-07-13

**Soundness:** 3
**Presentation:** 3
**Contribution:** 3
**Rating:** 5
**Confidence:** 3

**Summary:**

The paper introduces MC-DiT, a training paradigm for Diffusion Transformers (DiT) in the field of generative diffusion models for image generation.  By utilizing the proposed clean-to-clean mask-reconstruction approach, the model can better leverage contextual information at different noise variances.

**Strengths:**

- The paper provides a perspective on the limitations of noisy-to-noisy masked reconstruction, supported by theoretical insight and empirical analysis.
- The method is overall reasonable.
- The performance seems good.

**Weaknesses:**

- Will the additional two branches of DiT decoders increase the training overhead compared with other baseline methods? How about the training cost of each iteration compared with baselines?
- Comparing with MDT-XL / 2-, the improvements of MC-DiT-XL / 2-G seem to be marginal.
- How is the natural information measured in Fig. 1?
- Will the code be released?

**Questions:**

See the weakness part.
Besides, why is the IS of MC-DiT-XL / 2 much higher than other competitors?

---

> ### Author Rebuttal · Authors · 2024-08-07
>
> ### Q1: Training overhead of extra branches.
> With the additional two branches, the training cost of MC-DiT is a little bit higher than MaskDiT and MDT. As shown in below Table, the MC-DiT-XL/2 has more parameters, since the EMA branches introduce additional 56M parameters and these additional parameters only accounts for 7.6\%. The FLOPs and training speed of  MC-DiT-XL/2 are lower than those of MaskDiT-XL/2. The MDT-XL/2 has higher FLOPs and lower training speed than MC-DiT-XL/2 due to the difference of mask ratio (50% in MC-DiT and 30% in MDT).
> In a word, the additional overhead of MC-DiT is relatively small (7.6$\%$ parameters and $8\%$ FLOPs), but the FID performance improvement is significant.
> |$256\times256$|Params|FLOPs|Memory|Speed|FID|
> |:-------:|:-------:|:-------:|:-------:|:-------:|:-------:|
> |MDT-XL/2|742M|28G|20G|1.22|6.23|
> |MaskDiT-XL/2|730M|24G|18G|3.09|5.69|
> |MC-DiT-XL/2|786M|26G|20G|1.41|4.14|
>
> |$512\times512$|Params|FLOPs|Memory|Speed|FID|
> |:-------:|:-------:|:-------:|:-------:|:-------:|:-------:|
> |MDT-XL/2|742M|64G|28G|0.83|-|
> |MaskDiT-XL/2|730M|56G|24G|1.98|10.79|
> |MC-DiT-XL/2|786M|60G|27G|1.05|9.30|
>
> ### Q2: Improvements of MC-DiT-XL/2-G over MDT-XL/2-G.
> In $256\times 256$ image generation, MDT-XL/2-G and MC-DiT-XL/2-G obtain FID scores of 1.79 and 1.78, respectively. However, MDT-XL/2-G is trained for 6500K iterations, while our MC-DiT-XL/2-G only requires 2500K iterations, less than 40\% of MDT-XL/2-G. For a fair comparison, we train MDT-XL/2 and MDT-XL/2-G under similar 2500K training iterations for evaluations and report the results in below Table. MC-DiT-XL/2 and MC-DiT-XL/2-G are shown to outperform the retrained MDT-XL/2 and MDT-XL/2-G with evident margins. This results further demonstrates that our MC-DiT can achieves superior performance with much fewer training iterations.
> |Methods|Iterations|FID|sFID|IS|Prec.|Rec.|
> |:-------:|:-------:|:-------:|:-------:|:-------:|:-------:|:-------:|
> |MDT-XL / 2|2500K|7.41|4.95|121.22|0.72|0.64|
> |MaskDiT-XL / 2|2500K|5.69|10.34|177.99|0.74|0.60|
> |SD-DiT-XL / 2 |2400K|7.21|5.17|144.68|0.72|0.61|
> |MC-DiT-XL / 2|2500K|4.14|6.96|309.69|0.83|0.62|
> |MDT-XL / 2-G|2500K|2.15|4.52|249.27|0.82|0.58|
> |MaskDiT-XL / 2-G|2500K|2.28|5.67|276.56|0.80|0.61|
> |MC-DiT-XL / 2-G|2500K|1.78|4.87|290.17|0.81|0.62|
>
> ### Q3: Measure of Mutual information in Fig. 1.
> In Fig. 1, we measure the mutual information between unmasked and masked patches in different vanilla and generated images. Specifically, given a vanilla image, we patchify it via a tokenizer and apply the mask to obtain the masked and unmasked patches. Noise at different scales is add to these patches. For 'vanilla clean and noisy images' in Fig. 1, these unmasked patches and masked patches are then reshaped to one-dimensional tensors and the mutual information is calculated based on these tensors. For 'generated images' in Fig. 1, unmasked patches are sent to corresponding models for denoising and we use output patches for calculation.
>
> ### Q4: Will the code be released?
> We have provided the source codes in the supplementary material for review. The overall source codes and pretrained models will be released upon acceptance.

---

> > ### Comment · Reviewer_gqjf · 2024-08-12
> >
> > Thanks for your reply. I do not have further concerns.

---

> > > ### Author Response · Authors · 2024-08-13
> > >
> > > Thank you very much for your careful and insightful feedback, which has greatly improved the paper. We greatly appreciate your insights and are pleased to hear that all of your concerns have been addressed. Many thanks for your comments again.

---

### Official Review · Reviewer_TpTX · 2024-07-13

**Soundness:** 3
**Presentation:** 2
**Contribution:** 3
**Rating:** 5
**Confidence:** 3

**Summary:**

This paper observes that reconstructing masked noisy patches from unmasked noisy patches harms contextual information extraction during the training of DiT and then proposes a novel training paradigm named MC-DiT with clean-to-clean mask-reconstruction. Two EMA branches of DiT decoders are designed to avoid model collapse.

**Strengths:**

1. The manuscript adequately puts forth a number of propositions and commendably supports these with ample evidence and rigorous demonstrations, fostering a robust intellectual foundation for their arguments.
2. The authors' perspective on applying a noisy-to-noisy mask reconstruction approach is convincingly articulated.

**Weaknesses:**

1. The presentation of generated images for visualization is rather limited in quantity, necessitating an expansion to adequately illustrate the diversity and quality of the results. It is suggested to present generated results with the resolution of $512\times 512$. This paper only provides visual results in Figure 5 with the resolution of $256\times 256$ and it also claims superiority on $512\times 512$ image generation.
2. Lack of experiment details about training time, inference time and memory usage.

**Questions:**

1. Please clarify the reason why two extra EMA branches can address model collapse.
2. It is suggested to provide visual comparisons compared with other SOTA methods.
3. Investigating the impact of classifier-free guidance is recommended since it can improve the performance of many baselines such as ADM, DiT, and MaskDiT.

**Limitations:**

They provide accurate limitations in the conclusion section.

---

> ### Author Rebuttal · Authors · 2024-08-07
>
> ### Q1: Visual quality comparison.
> In Figures~R-1 and R-2(a) in the global rebuttal file, we further provide various $256\times 256$ and $512\times 512$ images generated by our MC-DiT and compare with SOTA methods MaskDiT and MDT. Our generated images are more realistic and have more consistent textual structure than MaskDiT and MDT. For example, images of the 'school bus' and 'custard apple' in Figure R-1(a) exhibit different styles, while the details appear very realistic. Moreover, images of 'hammer' generated by
>  MaskDiT and MDT have incomplete structure, while our MC-DiT generates images with more complete structures. The same thing happened in $512\times 512$ images in Figure R-1(b), various images (e.g. 'dog', 'fox' and 'penguin') exhibit rich details and realistic styles, validating the effectiveness of our MC-DiT.
> ### Q2: Details about training time, inference time, and memory usage.
> Below table elaborates the implementation details for training. We leverage two types of GPUs for training and summarize the training time, inference time and memory usage.
> |Setting|MC-DiT-B/2|MC-DiT-XL/2|MC-DiT-XL/2|
> |:-:|:-:|:-:|:-:|
> |Resolution|$256\times256$|$256\times256$|$512\times512$|
> |Training Time|50h|586h|623h|
> |Inference Time (50K images)|12h|8h|15.2h|
> |GPUs|2$\times$RTX-3090 GPUs|4$\times$V100GPUs|4$\times$V100GPUs|
> |Batch Size|256$\times$2|256$\times$4|128$\times$4|
> |Memory Usage per GPU|17GB|20GB|27GB|
>
> Besides, we compare the training cost (parameters, FLOPs, memory used and training speed) on 4$\times$V100 GPUs in below Tables, where training speed denotes the number of iterations per second. The training speed of MC-DiT is a little bit slower than other methods due to two EMA branches. However, the inference speed of MC-DiT is similar to MaskDiT, since two EMA branches are removed during inference. The additional overhead of MC-DiT is relatively small (7.6$\%$ parameters and $8\%$ FLOPs), but the FID performance improvement is significant.
> |$256\times256$|Params|FLOPs|Memory|Speed|FID|
> |:-------:|:-------:|:-------:|:-------:|:-------:|:-------:|
> |MDT-XL/2|742M|28G|20G|1.22|6.23|
> |MaskDiT-XL/2|730M|24G|18G|3.09|5.69|
> |MC-DiT-XL/2|786M|26G|20G|1.41|4.14|
>
> |$512\times512$|Params|FLOPs|Memory|Speed|FID|
> |:-------:|:-------:|:-------:|:-------:|:-------:|:-------:|
> |MDT-XL/2|742M|64G|28G|0.83|-|
> |MaskDiT-XL/2|730M|56G|24G|1.98|10.79|
> |MC-DiT-XL/2|786M|60G|27G|1.05|9.30|
>
> ### Q3: Two extra EMA branches for addressing model collapse.
> The modal collapse occurs when the main branch only considers clean-to-clean mask-reconstruction for masked clean patches but ignores the denoising of unmasked noisy patches. We propose two EMA branches to balance the two tasks for the main branch. We use the noisy EMA branch to realize noisy-to-clean mapping for denoising, and the clean EMA branch to realize clean-to-clean mapping for mask-reconstruction (mask ratio is 0\%). The two EMA branches constrain the output of the main branch (minimize the MSE loss between the outputs of the main branches and EMA branches) via three hyper-parameters, which leads to the balance on the denoising task and clean-to clean mask-reconstruction task.
>
> To verify this, we report in below Table the FID score of the main branch with noisy and clean patch inputted only. The result of the main branch with unmasked noisy patches only is higher than that of masked clean patches, indicating the modal collapse problem. With noisy and clean branches, the FID score of the main branches decline distinctly, validating the effectiveness of the EMA branches.
> |Branches|FID|
> |:-------:|:-------:|
> |Min Branch|22.10|
> |w Noisy Branch|19.26|
> |w Clean Branch|18.88|
> |Main Branch (unmasked noisy patch only)|25.72|
> |Main Branch (masked clean patch only)|23.69|
> |Main Branch (unmasked noisy patch only) w Noisy Branch|19.84|
> |Main Branch (unmasked noisy patch only) w Clean Branch|19.57|
>
> ### Impact of classifier-free guidance.
> We have reported the $256\times256$ image generation results with classifier-free guidance (CFG) in Table 1 in the manuscript, indicating the superior performance of MC-DiT. Besides, we supplement the $512\times512$ results in the below Table. CFG benefits our MC-DiT with a district margin (4.14 vs 1.78 for $256\times 256$ and 9.30 vs 2.03 for $512\times 512$).
> |Methods|FID|sFID|IS|Prec.|Rec.|
> |:-------:|:-------:|:-------:|:-------:|:-------:|:-------:|
> |DiT-XL / 2|12.03|7.12|105.25|0.75|0.64|
> |MaskDiT-XL / 2|10.79|13.41|145.08|0.74|0.56|
> |MC-DiT-XL / 2|9.30|6.28|179.58|0.76|0.53|
> |DiT-XL / 2-G|3.04|5.02|240.82|0.84|0.54|
> |MaskDiT-XL / 2-G|2.50|5.10|256.27|0.83|0.56|
> |MC-DiT-XL / 2-G|2.03|4.87|272.19|0.84|0.56|

---

> > ### Author Response · Authors · 2024-08-12
> >
> > Thank you very much for the time and effort you have dedicated to reviewing our manuscript.  We believe we have addressed all the concerns you raised in your review and expect your feedback sincerely.
> > Thank you once again for your attention.

---

> > > ### Comment · Reviewer_TpTX · 2024-08-13
> > >
> > > Thank you for your rebuttal. Most of my concerns are addressed. I think the experiments shown in the original paper are not sufficient and enough and it would be better to include the results in the rebuttal. I will change the rating.

---

> > > > ### Author Response · Authors · 2024-08-13
> > > >
> > > > Thank you very much for recognizing the improvements made to the paper and re-assessing the rating. We sincerely appreciate your thoughtful evaluation and the time you dedicated to reevaluating our work! We will definitely include the results provided in the rebuttal in the final version of our paper.

---

### Official Review · Reviewer_cs4r · 2024-07-25

**Soundness:** 3
**Presentation:** 4
**Contribution:** 3
**Rating:** 5
**Confidence:** 3

**Summary:**

This paper introduces MC-DiT, a novel training paradigm for Diffusion Transformers (DiT) in image generation. It addresses the limitations of current masked-reconstruction strategies, which fail to effectively extract **contextual information** due to noisy-to-noisy reconstruction. MC-DiT employs clean-to-clean reconstruction, allowing for better contextual information utilization during diffusion denoising. The authors also design dual decoder branches to prevent model collapse. Theoretical and empirical analyses validate their approach, and experiments on the ImageNet dataset show that MC-DiT achieves state-of-the-art performance in both unconditional and conditional image generation tasks.

**Strengths:**

1.The introduction of the MC-DiT paradigm, which utilizes clean-to-clean mask-reconstruction, represents a novel approach that addresses the limitations of existing methods in extracting contextual information.

2.The authors provide a thorough theoretical and empirical analysis, particularly focusing on mutual information, which strengthens the validity of their claims.

3. The proposed MC-DiT achieves superior results in both unconditional and conditional image generation tasks, as demonstrated by the state-of-the-art FID scores on the ImageNet dataset.

**Weaknesses:**

1. The paper primarily focuses on image generation using the ImageNet dataset. It remains to be seen how well the approach generalizes to other domains or datasets with different characteristics.

2. The authors should clearly elaborate on the differences between MC-DiT and other masked diffusion transformers (such as MaskGiT,SD-DiT, and MaskDiT).

**Questions:**

The proposed method may still require large computational resources due to the dual-branch decoder design and the clean-to-clean reconstruction process. How to accelerate it?

**Limitations:**

The authors acknowledge that the training and inference speeds need improvement.

---

> ### Author Rebuttal · Authors · 2024-08-07
>
> ### Q1: Generalization to other domains or datasets.
> We adopt the ImageNet dataset in the experiments for a fair comparison, since MaskDiT, SD-DiT and MDT are all evaluated on the ImageNet dataset. In fact, our MC-DiT can be generalized to different domains or datasets for improved image generation due to the fact that it can extract contextual information from arbitrary images. Table below compares the performance of MaskDiT and MC-DiT on the CIFAR-10 and CelebA (that collected for face recognition) datasets. Due to time limit, we train both MaskDiT and MC-DiT for 200K iterations. Experimental results show that MC-DiT outperforms MaskDiT on both datasets.
> |Cifar10|FID|
> |:--:|:--:|
> |MaskDiT-B / 2|11.52|
> |MC-DiT-B / 2|9.28|
>
> |CelebA|FID|
> |:--:|:--:|
> |MaskDiT-B / 2|7.14|
> |MC-DiT-B / 2|5.36|
> ### Q2: Difference between MC-DiT and other masked DiTs.
> In summary, we propose clean-to-clean reconstruction for MC-DiT achieve enhanced contextual information extraction, while existing methods (MDT, SD-DiT, and MaskDiT) are limited in contextual information extraction using noisy-to-clean and noisy-to-noisy reconstruction. Specifically, in Section 3.2 in the manuscript, we first demonstrate the problem of limited ability in contextual information extraction for previous methods (MDT, SD-DiT, and MaskDiT), since they apply noisy patches reconstruction task into the training process of DiT. They could suffer from weak ability of contextual information extraction when the noise is large. Based on this finding, our MC-DiT leverages clean patches for mask-reconstruction task to model contextual information effectively. We insert masked clean patches into the unmasked noisy features for reconstruction to exploit clean contextual information for denoising unmasked patches at arbitrary noise scales. Experimental results demonstrate that our MC-DiT can extract sufficient contextual information and achieve superior performance.
> ### Q3: Computational resources and acceleration.
> We report the resource consumption of MC-DiT, MDT, and MaskDiT in Table R-1(a) in the one-page pdf. Compared with MaskDiT, our MC-DiT sligtly increases the parameters and FLOPs by 7.6\% and 8\% but yields obvious FID gains of 1.5 and 1.4 for $256\times 256$ and $512\times 512$ image generation. MC-DiT could be potentially accelerated by employing more layers for DiT encoder but less layers for DiT decoder to decrease the overhead of DiT decoder.
> |$256\times256$|Params|FLOPs|Memory|Speed|FID|
> |:-------:|:-------:|:-------:|:-------:|:-------:|:-------:|
> |MDT-XL/2|742M|28G|20G|1.22|6.23|
> |MaskDiT-XL/2|730M|24G|18G|3.09|5.69|
> |MC-DiT-XL/2|786M|26G|20G|1.41|4.14|
>
> |$512\times512$|Params|FLOPs|Memory|Speed|FID|
> |:-------:|:-------:|:-------:|:-------:|:-------:|:-------:|
> |MDT-XL/2|742M|64G|28G|0.83|-|
> |MaskDiT-XL/2|730M|56G|24G|1.98|10.79|
> |MC-DiT-XL/2|786M|60G|27G|1.05|9.30|

---

> > ### Comment · Reviewer_cs4r · 2024-08-11
> >
> > Thanks for the rebuttal, and I would like to keep my original rating.

---

> > > ### Author Response · Authors · 2024-08-12
> > >
> > > Thank you for your thoughtful feedback and for taking the time to review our paper. We are glad that our rebuttal has addressed all of your concerns. Many thanks for your comments again.

---

### Official Review · Reviewer_Ngpd · 2024-07-26

**Soundness:** 4
**Presentation:** 4
**Contribution:** 4
**Rating:** 5
**Confidence:** 4

**Summary:**

The paper introduces a novel training paradigm for Diffusion Transformers (DiT) in the context of generative diffusion models for image generation. The authors propose MC-DiT, which focuses on enhancing contextual information extraction by reconstructing clean unmasked patches from clean masked patches, as opposed to the traditional noisy-to-noisy reconstruction. The method employs two complementary branches of DiT decoders to balance the use of noisy and clean patches, preventing model collapse.

**Strengths:**

1.	The paper presents a new insight into the use of clean-to-clean reconstruction for learning contextual information in masked diffusion models, which is a significant departure from traditional noisy-to-noisy reconstruction methods.
2.	The authors provide a theoretical analysis of mutual information between unmasked and masked patches, demonstrating the limitations of existing methods and the benefits of their proposed approach.
3.	The introduction of two complementary DiT decoder branches to prevent model collapse is a thoughtful addition that addresses a common issue in such models.
4.	The paper reports state-of-the-art results in terms of FID scores and IS scores, indicating that the proposed MC-DiT is highly competitive with existing methods.

**Weaknesses:**

1.	The proposed MC-DiT model may be more complex than necessary, which could potentially hinder its adoption and implementation in practical applications.
2.	The paper acknowledges that the training and inference speed of MC-DiT needs to be improved, which suggests that the current approach may have efficiency issues. The authors should provide specific comparisons to demonstrate that these efficiency sacrifices are worth the performance gains.
3.	The paper could benefit from a more detailed comparative analysis with other state-of-the-art methods, including feature visualization, to better understand the advantages of MC-DiT.
4.	Can the author explain whether this specific context information is pixel-wise information or semantic information? And their role in the overall framework?
5.	Ablation experiments can be further supplemented and improved. For example, the hyperparameters in Tab.5 can be further observed to have an impact. The current scaling still has some ambiguity.

**Questions:**

Please refer to the Weakness Section.

**Limitations:**

The authors have mentioned the limitations in this paper.

---

> ### Author Rebuttal · Authors · 2024-08-07
>
> ### Q1: Complexity.
> Our MC-DiT has the same main branch and training objective as existing methods like MaskDiT, MDT, and SD-DiT. The additional complexity of MC-DiT lies on the extra two EMA branches and unmasked tuning.
>
> 1) The extra two branches increases only 7.6\% parameters and 8\% FLOPs, as shown in Figure R-1(a) in the one-page pdf. Thus, MC-DiT is approximately similar as MaskDiT, MDT and SD-DiT in training cost. Moreover, during inference, the two extra EMA branches are dropped without causing extra complexity, and MC-DiT has the same architecture as DiT in the inference stage.
> 2) Unmasked tuning can reduce the training-inference discrepancy, as demonstrated by MaskDiT and is adopted in our MC-DiT. However, we can remove unmasked tuning to reduce the complexity at little loss on FID. Table below shows that FID will increase by only 0.41 for MC-DiT by removing unmasked tuning.
> |Strategy|Iterations|FID|
> |:---:|:---:|:---:|
> |MaskDiT-XL/2 w unmasked tuning|1300K|12.15|
> |MC-DiT-XL/2 w unmasked tuning|1300K|7.92|
> |MC-DiT-XL/2 w/o unmasked tuning|1300K|8.33|
>
> ### Q2: Training and inference speed.
> First of all, we have to clarify that, in the limitations, we mean to improve the inference speed of diffusion models rather than the inference speed of MC-DiT compared to existing DiT methods. In fact, MC-DiT has the same inference speed as MaskDiT, since two extra branches are removed during inference. Regarding training speed, MC-DiT is slower than MaskDiT due to the two extra EMA branches. However, MC-DiT yields an evidently lower FID scores (i.e., 1.5 for $256\times 256$ image generation and 1.4 for FID for $512\times512$ image generation) than MaskDiT.
> ### Q3: More detailed comparative analysis
> We further provide visualization of the feature maps extracted by MC-DiT and MaskDiT at different noise scale on CIFAR-10 in Figure R-2(b) in the one-page pdf. A larger noise variance denotes the noise with large scale. Our MC-DiT extracts proper shape for various noise scale, while the features extracted by MaskDiT are messy in the large noise scale. This further proves the motivation and effectiveness of our paper that clean-to-clean mask reconstruction promotes learning sufficient contextual information.
> ### Q4: Specific context information.
> The context information is semantic information that denotes the relationship between current patches and other patches. For example, the patches of legs are correlated with patches of the body in a cat image.  In fact, MAE employs mask-reconstruction for contextual information extraction from the clean image. In this paper, we introduce mask-reconstruction into the denoising process to extract contextual information. Specifically, the masked clean patches contain contextual information about clean patches target. Using masked clean patches to reconstruct target clean patches helps the model to understand the shape and context.
> ###Q5: Ablation on hyperparameters.
> Following MaskDiT, we select 0.01, 0.1, and 1.0 as the scaling values of three hyperparameters and supplement various values for ablation study. Below tables evaluate various values of the three hyperparameters and we find that the best FID is still obtained when $\lambda_1=0.1$, $\lambda_2=0.1$, and $\lambda_3=0.05$.
> |$\lambda_1$|0|0.01|0.03|0.05|0.07|0.09|0.1|0.3|0.5|0.7|0.9|
> |:---:|:---:|:---:|:---:|:---:|:---:|:---:|:---:|:---:|:---:|:---:|:---:|
> |FID|43.23|40.99|39.23|38.44|37.95|36.53|35.20|35.98|36.74|36.91|37.52|38.97|
>
> |$\lambda_2$|0|0.01|0.03|0.05|0.07|0.09|0.1|0.3|0.5|0.7|0.9|
> |:---:|:---:|:---:|:---:|:---:|:---:|:---:|:---:|:---:|:---:|:---:|:---:|
> |FID|38.83|36.15|36.02|36.46|35.99|36.07|35.20|35.34|36.18|37.26|35.98|37.54|
>
> |$\lambda_3$|0|0.01|0.03|0.05|0.07|0.09|0.1|0.3|0.5|0.7|0.9|
> |:---:|:---:|:---:|:---:|:---:|:---:|:---:|:---:|:---:|:---:|:---:|:---:|
> |FID|37.77|37.25|36.63|35.20|36.07|37.93|35.46|35.88|37.26|36.19|37.40|36.35|

---

### Official Review · Reviewer_RKJM · 2024-07-28

**Soundness:** 3
**Presentation:** 3
**Contribution:** 2
**Rating:** 6
**Confidence:** 3

**Summary:**

In this work, the authors reveal the issues of Diffusion transformers of having semantic inconsistency as they fail to learn the contextual information. Based on their theoretical analysis, they proposed a novel training paradigm to fully learn contextual information with clean-to-clean mask reconstruction. The paper is well organised and written.

**Strengths:**

The authors have a comprehensive understanding of issues and the state-of-the-art. In terms of originality and quality, the work is technically sound in general. The analysis and written are clear in general.

**Weaknesses:**

Please see the list of questions for improvement and clarification on some of the aspects.

**Questions:**

1. The authors gave a thorough analysis on the issues of diffusion transformers in section 3.  However, the motivation for the proposed MC-DiT to solve the issues is not very clear.
2. The steps mentioned in section 3.3 are not so clear and probably are not cohesive with Figure 2. For instance, it mentioned as ‘the unmasked noisy patches x_t^{1} are fed into the DiT encoder for extraction’, but it seems those unmasked noisy patches go to the DiT decoder (?). I might be better to put the denotations on Figure 2 as well to guide readers.
3. In Table 1, the results on using classier-free guidance were reported for ImageNet-256x256 generation. However, when it comes to the ImageNet-512x512 generation, they are ignored and not reported. Any particular reason behind?

**Limitations:**

The authors mentioned that the training speed and inference speed still need to be improved and a possible future mitigation on the issue.

---

> ### Author Rebuttal · Authors · 2024-08-07
>
> ### Q1: Motivation for MC-DiT
> In Section 3 of the manuscript, sufficient analysis is provided to claim that reconstructing masked noisy patches from unmasked noisy patches is insufficient for contextual information extraction. In details, the information used in noisy-to-noisy patch reconstruction only lie in $\mathcal{I}(x_0^1;x_t^2)$ or $\mathcal{I}(x_t^1;x_t^2)$, which are less than $\mathcal{I}(x_0^1;x_0^2)$. Thus, our motivation is to directly model $\mathcal{I}(x_0^1;x_0^2)$ via the clean-to-clean patch reconstruction, which reduces the impact of the noise and learns sufficient contextual information. This is realized by inserting masked clean patches into the unmasked noisy features for unmasked clean patches reconstruction. The contextual information flows from the masked clean patches to the unmasked clean patches, which equals $\mathcal{I}(x_0^1;x_0^2)$.
> ### Q2: Denotations on Figure 2.
> Thank you for the comment and we will put the denotations on Figure 2 in the revised version. Here, we would like to explain the reason that `$x_t^{1}$ are fed into the DiT decoder for extraction as below.
>
> In the main branch, the unmasked noisy patches are fed into the DiT encoder, while all the noisy patches are directly inserted into the EMA DiT decoder to avoid modal collapse, as shown in the Figure 2 in the manuscript. The reasons are on the two folds: (1) efficient. Only apply DiT decoder for EMA branches leads to small extra parameters and fast inference speed, while EMA DiT encoder slows down the entire EMA branches. (2) effective. The DiT decoder is trained to extract masked clean images patches in the main branch. Thus, directly apply image patches as the input of EMA DiT decoder does not lead to poor denoising results. As shown in below Table , applying EMA DiT encoder introduces extra 669M parameters, while FID score only decreases 1.35. Thus, to balance the parameters and performance, we select DiT decoder in the EMA branches.
> |Branch|Params|FID|
> |:--:|:--:|:--:|
> |DiT Decoder|56M|18.88|
> |DiT Decoder+DiT Encoder|725M|17.53|
>
> ### Q3: Classier-free guidance for ImageNet-512x512 generation.
> According to your suggestion, we provide the results for ImageNet-512x512 generation with classier-free guidance in Table R-1 (b) in the one-page pdf. Compared with MaskDiT, our MC-DiT is consistent to reduce FID by 0.47, demonstrating its effectiveness.

---

### Official Review · Reviewer_YSTZ · 2024-07-29

**Soundness:** 2
**Presentation:** 3
**Contribution:** 3
**Rating:** 5
**Confidence:** 4

**Summary:**

This paper proposes a training strategy for diffusion transformers that fully learns contextual information by introducing clean to clean mask reconstruction during training, and designs complementary DiT decoder branches as well as corresponding supervisory losses to avoid the problem of model collapse, giving theoretical and experimental validation.

**Strengths:**

1.	Sufficient theoretical analysis
2.	The overall writing of the paper is logically clear

**Weaknesses:**

1.	There are errors in the description of parts of the paper, e.g., x1 in lines 107 and 109 of the introductory section of Masked AutoEncoders is described as masked and unmasked, respectively.
2.	Visualization of experimental results is indeed missing, and only quantitative experimental results exist in the body of the paper.
3.	Using the training strategy in the paper, although it can improve the results, it is not possible to conclude the size of the contribution of the training strategy to the final experimental results, as parameter tuning is still required in the testing phase.

**Questions:**

1.	Why not release the qualitative results as proof of the effectiveness of the strategy?
2.	How much does parameter tuning in the final testing phase affect the degree of merit of the final result? How can it be shown that it is the training strategy that is at work and not the parameter tuning?

**Limitations:**

The methodology proposed in the paper needs to be fine-tuned to the task during the inference phase, otherwise good results may not always occur. Further, it can be found that similar strategies that utilize unknown information in the inference phase for training can have application limitations, which are not conducive to extending and applying the strategy to tasks that do not have access to potentially clear images and information.

---

> ### Author Rebuttal · Authors · 2024-08-07
>
> ### Q1: Writing error in line 107.
> We thank the reviewer for point out the writing error. The unmasked patches $x_1$ and masked patches $x_2$ in line 107 are corrected to $x_1=x[m]$ and $x_2=x[1-m]$.
> ### Q2: Visualization results.
> We have provided visualization results of generated $256\times 256$ images in Figure 5 in the supplementary materials. Figure 5 shows that images generated by MC-DiT achieve vivid details and diverse styles. In this rebuttal, we further provide more visualization results for both generated $256 \times 256$ and $512\times 512$ images in the Figure R-1 in the global rebuttal file. It demonstrates that images of the 'school bus' and 'custard apple' in Figure R-1(a) exhibit different styles, while the details appear very realistic. The same thing happened in $512\times 512$ images in Figure R-1(b), various images (e.g. 'dog', 'fox' and 'penguin') exhibit rich details and realistic styles, validating the effectiveness of our MC-DiT.
> ### Q3: Parameter tuning required for testing.
> We would like to emphasize that it is the enhanced ability of contextual information extraction by the clean-to-clean patch reconstruction rather than parameter tuning contributes to the superior performance of our MC-DiT. Existing masked diffusion models like MaskDiT and SD-DiT require parameter (unmasked) tuning to decrease the training-inference discrepancy. Different from these models, MC-DiT leverages clean-to-clean patch reconstruction to enhance the contextual information extraction for generation beyond the unmasked tuning to decrease the training-inference discrepancy. Experimental results further demonstrate the effectiveness of the clean-to-clean patch reconstruction in MC-DiT.
>
> **Experimental evaluations.** To validate the effectiveness of the clean-to-clean patch reconstruction, we compare MaskDiT-XL/2 (with unmasked tuning) and MC-DiT-B/2 with and without unmasked tuning. All the models are trained for 1300K iterations. As reported in below Table, both MC-DiT-B/2 with and without unmasked tuning outperform MaskDiT-XL/2 by evident margins of 4.27\% and 3.82\% in FID. By contrast, masked tuning only leads to a reduction of 0.41 in FID. These results demonstrate that the performance gain of our MC-DiT mainly comes from the enhanced ability of contextual information extraction by clean-to-clean patch reconstruction rather than unmasked tuning.
> |Strategy|Iterations|FID|
> |:---:|:---:|:---:|
> |MaskDiT-XL/2 w unmasked tuning|1300K|12.15|
> |MC-DiT-XL/2 w unmasked tuning|1300K|7.92|
> |MC-DiT-XL/2 w/o unmasked tuning|1300K|8.33|

---

> > ### Author Response · Authors · 2024-08-13
> >
> > Thank you very much for the time and effort you have dedicated to reviewing our manuscript. We believe we have addressed all the concerns you raised in your review and expect your feedback sincerely. Thank you once again for your attention.

---

> ### Comment · Reviewer_YSTZ · 2024-08-14
>
> Thank you for the author's response. I suggest including the visualization results and ablation experiments in the final version of the paper. I have raised my score accordingly.

---

> > ### Author Response · Authors · 2024-08-14
> >
> > Thank you very much for your comprehensive evaluation and the time you dedicated to reevaluating our work! We will include the experiments and visualization provided in the rebuttal in the final version of our paper. Many thanks for your suggestions again!

---

### Official Review · Reviewer_ZkfZ · 2024-07-29

**Soundness:** 3
**Presentation:** 2
**Contribution:** 3
**Rating:** 7
**Confidence:** 4

**Summary:**

This paper critiques previous masked-reconstruction strategies in DiT training for their poor contextual information extraction, attributing this to noisy-to-noisy reconstruction. The authors theoretically and empirically validate that this approach limits mutual information between unmasked and masked patches. To address this, they propose a new training paradigm, MC-DiT, which uses clean-to-clean mask-reconstruction combined with diffusion denoising at varying noise levels. To prevent model collapse, they design two complementary DiT decoder branches to balance the reliance on noisy and clean patches. Model collapse would happen in this context due to excessive reliance on clean patches for reconstruction, leading to insufficient utilization of noisy patches and imbalanced training. Extensive experiments on the ImageNet dataset show that MC-DiT achieves state-of-the-art performance in both unconditional and conditional image generation, with faster convergence.

**Strengths:**

- The paper motivates the need for their research in their introduction and is an interesting idea.
- The paper adds to the mathematical discussion surrounding image generation using diffusion using well understood mutual information metric prevalent in other areas of computer vision.
- Presents experimental evaluation, with section on reproducible details and supplementary materials.

**Weaknesses:**

- While reading the article, there are many questions that arise which effect the reading experience of the article.
- The main weakness of the paper is that at many occasions claims are made which are intuitive, but they are attributed to be implied from an equation / proposition which do not (at least not immediately) show the claim to be true. Look at questions for more details.
- Some experiment details are unclear (in questions).
- Table 1 is a bit difficult to read here with the number of methods and it is not obvious how the horizontal lines are drawn, i.e. what makes them different from other quadrants. I think there is enough space for a column or two to add a bit more detail instead of adding them all to the name of the method.
- Figure 3 (a) is used to showcase speed of convergence. However, I think the distinction between convergence and a convergence to lower loss should be made. All 3 lines more or less flatten at the same time, you could actually argue the red and orange line are flattening faster. I agree the blue line is lower, but that does not mean it has converged faster, only converged to a lower loss. This also leads to a second point, a lower loss here does not necessarily mean a more performant model. As you notice in your own experiments, you require fine-tuning to make the output desirable. Therefore, I disagree that the model converges faster on the whole.

**Questions:**

- Where can I see: Line 38 "Despite superior performance over vanilla DiT, they are deficient in exploiting contextual information by neglecting different noise scales in different steps of diffusion process." Is there a citation which discusses this is important, or this concluded from your Figure 1 and table of results Table 1?
- It is not obvious from equation 5 and 6 that in Line 163: "With the growth of $t$, the KL divergence terms in (5) and (6) increase due to larger noise perturbation on $x^1_0$ and $x^2_0$" should be true. This can be understood intuitively, but there is no "decay" term with respect to $t$ in these equations to suggest that. Can this be formalised with respect to strength of the gaussian noise $n$. Also, I do realise that due to non negativity of KL divergence the two expectation terms subtract from it to make the mutual information smaller, but I so not see how this is true let's say between t and t+1.
- Line 216: how are the 2 branches of DiT trained in the EMA fashion here (student teacher or are do they also collect gradients)?
- Line 203: "$\mathcal{I}(x^1_0; x^2_0)$ is much higher", the **much** part is not clear from Proposition 2.
- Figure 3, is the training loss that is logged for all the models the same? i.e. $\mathcal{L}_{\text{clean}}$? or for your method is it the composite loss?
- Related to Figure 3, when we talk about speed of convergence in terms of iterations it does not say anything about the wall clock time (or FLOPs or Memory) that an iteration takes. In this adapted method, we do x3 forward passes through the DiT decoder, therefore how do the wall clock times (or FLOPs or Memory) compare? From a practical standpoint, this should be clear. Hypothetically, do you also expect the other methods to make up the difference in performance if they were trained for a proportional time longer.
- Figure 3 a and b, why is the plot only shown for different number of total iterations?
- Figure 3b, are these metrics calculated before or after fine-tuning for your method?
- Line 309 in Limitations. Why does the inference speed need to be improved? Is the model inferred differently and requires more steps?

*Minor Typos*
- Line 13: MDT mentioned before it is defined, although this clear from the citation
- Equation 1: it was not clear $\mathcal{L}_{\text{asym}}$ was defined as the expectation term, this lead to some confusion in the discussion later in Proposition 3.
- Proposition 2, Equation 6 should end with a full stop.
- Equation 8 9, 10, 11: brackets are not matched, duplicate sencond closing brackets?

**Limitations:**

Questions listed above.

---

> ### Author Rebuttal · Authors · 2024-08-07
>
> ### Q1:Claim in Line 38
> We provide both theoretical and empirical evidences in Proposition~2 and Figure 1(a) in the manuscript to support the claim. We consider the mutual information $\mathcal{I}(x_0^1;x_0^2)$ between unmasked patches $x_0^1$ and masked patches $x_0^2$ as the contextual information.
> Proposition 2 points out that previous methods only learns $\mathcal{I}(x_0^1;x_t^2)$ and $\mathcal{I}(x_t^1;x_t^2)$, which is insufficient for $\mathcal{I}(x_0^1;x_0^2)$. Figure 1(a) demonstrates that the mutual information learned by previous methods (MaskDiT, DiT and MDT) all declines greatly during the noise scale increases, indicating that these methods struggle to learn contextual information under different noise scales.
> ### Q2:Equations(5) and (6) and Claim in Line 163.
> The details of the KL divergence in the Proposition 2 are as follows
> $$
> E_{p(x_0^2)}E_{p(x_t^2|x_0^2)}E_{p(x_0^1|x_0^2)}\log \left[\frac{p(x_0^1|x_t^2)}{p(x_0^1|x_0^2)}\right] \\
> =E_{p(x_0^2)}E_{p(x_t^2|x_0^2)}E_{p(x_0^1|x_0^2)}\log \left[\frac{p(x_t^2|x_0^1)}{p(x_0^2|x_0^1)}\times \frac{p(x_t^2)}{p(x_0^2)}\right] \\
> \approx E_{p(x_0^2)}E_{p(x_t^2|x_0^2)}E_{p(x_0^1|x_0^2)} \log \left[\frac{p(x_0^2|x_0^1)+p(n|x_0^1)}{p(x_0^2|x_0^1)}\times \frac{p(x_0^2)+p(n)}{p(x_0^2)}\right] \\
> $$
> where $x_t^2=x_0^2+n$ with $n\in\mathcal{N}(0,t^2I)$. We approximate $p(x_t^2)\approx p(x_0^2)+p(n)$, since $p(x_t^2)$ is a Gaussion distribution with mean value $x_0^2$ and variance $t^2$. As $t$ increases, the KL divergence in Proposition 2 increases and the mutual information $\mathcal{I}(x_0^1;x_t^2)$ and $\mathcal{I}(x_t^1;x_t^2)$ achieve the larger difference with $\mathcal{I}(x_0^1;x_0^2)$.
> ### Q3:Training of the 2 branches of DiT
> The parameters of EMA decoders are initialized with those in the DiT decoders and are updated in the EMA fashion according to the parameters in DiT decoders. $\theta_{ema}=\alpha \times \theta_{ema}+(1-\alpha)\times \theta_{dec}$
> where $\alpha$ denotes the weight coefficient. The two decoders are updated only using the EMA method without using gradient updates.
> ### Q4:Much part of $\mathcal{I}(x_0^1;x_0^2)$
> We demonstrate in Proposition 2 that the difference between $\mathcal{I}(x_0^1;x_0^2)$ and $\mathcal{I}(x_0^1;x_t^2)$ lies in the KL divergence under various noise scale and further verify the theoretical results in Figure 1(a). Figure 1(a) shows that, with the growth of noise, the difference between $\mathcal{I}(x_0^1;x_0^2)$ (red line) and $\mathcal{I}(x_0^1;x_t^2)$ (gray and yellow lines) becomes larger. During the increase of noise variance from 0.0 to 1.0, the mutual information of MaskDiT and MDT decline 90% (1.95 vs 0.21), while mutual information of vanilla noisy image only decline 18% (2.07 vs 2.50). This decline refers to the KL divergence in Proposition 2.
> ### Q5:Training loss in Figure 3
> In Figure 3, $L_{clean}$ in Equation (8) is adopted as the training loss for all the models. Figure 3 shows that our MC-DiT converges to a lower value than other methods in terms of $L_{clean}$ and demonstrates the effectiveness of clean-to-clean mask reconstruction.
> ### Q6:Different numbers of total iterations in Figures 3a and b.
> To compare training performance, we train MaskDiT and DiT for 300K iterations in Figures 3a and b due to the substantial time and GPU resource overhead. We directly use the training curve of our MC-DiT trained for evaluations. MC-DiT is trained for 400K iterations for a fair comparison with other methods. In fact, we can find from the training curves for the first 300K iterations that, MC-DiT can obviously decrease the training loss and FID score in comparison to MaskDiT and DiT. This result implies the improvements of our MC-DiT.
> ### Q7:Metrics in Figure 3b.
> The FID reported in Figure 3(b) is calculated after unmasked tuning for MaskDiT and MC-DiT. It is consistent to the results reported in Table 3 in the manuscript.
> ### Q8:Inference speed in limitations
> We intend to improve the inference speed of diffusion model that requires multiple steps to achieve image generation, since MC-DiT is based on the diffusion model and shares the same architecture with MaskDiT. In fact, MC-DiT does not infer differently or requires more steps and yields the same time consumption in the inference stage, since the EMA branches and noisy target are removed during test.
> ### Q9:Evaluation of speed of convergence.
> We follow Figure 4 in SD-DiT to evaluate the speed of convergence for each iteration in Figure 3. Besides, as for the wall clock time, we report FLOPs and Memory used by MC-DiT in each iteration in Table R-1(a) in one page pdf. MC-DiT achieves 3x forward passes, thus the FLOPs and Memory are calculated via the summation of 3x forward process. Training speed of MC-DiT is slower than MaskDiT, leading to longer convergence time. However, MC-DiT converges to a lower loss and achieves superior FID score than MaskDiT. Meanwhile, due to the lightweight DiT decoder, MC-DiT achieves small extra FLOPs and Memory in the training stage.
> The loss curve of MaskDiT may not decrease due to the small decrease magnitude after 100K iterations. And the FID score of previous methods doesn't decline faster than MC-DiT according to the decrease magnitude in Figure 2(b) of the manuscripts.
> ### Q10:Speed of convergence shown in Figure 3(a).
> Although convergence to a lower loss is not bound to imply a faster convergence in terms of iterations, the primary focus of our analysis is the overall effectiveness of the model. The blue line can achieve a lower loss, despite similar iteration counts for flattening, highlights the model's efficiency in reaching a more optimal solution.
> Besides, the loss reported in Figure 3(a) denotes the MSE loss $\mathcal{L}_{clean}$. Thus, lower MSE loss means the generated clean patches are more similar to the ground-truth, indicating the performant model. Moreover, we report the results with and without unmasked tuning in Table. Our MC-DiT outperforms MaskDiT without unmasking tuning.

---

> ### Comment · Reviewer_ZkfZ · 2024-08-11
>
> Thank you for the clarifications and my questions have been sufficiently answered. I have updated my recommendation for the paper now.

---

> > ### Author Response · Authors · 2024-08-12
> >
> > Thank you very much for acknowledging the improvements made to the paper and reconsidering the score! We greatly appreciate your thoughtful evaluation and the time you took to reassess our work.

---

### Author Rebuttal · Authors · 2024-08-07

We thank all reviewers for their valuable comments. We appreciate the reviewers' recognition of our work, including **excellent motivation (ZkfZ, RKJM, Ngpd, cs4r and gqjf), reasonable method (gqjf), thorough theoretical analysis (cs4r, Ngpd, YSTZ), state-of-the-art results (Ngpd, gqjf, cs4r and ZkfZ), along with clear writing (YSTZ, RKJM)**. The reviewers also raised some issues, including training efficiency (TpTX,cs4r, RKJM and ZkfZ) and unmasked tuning (ZkfZ, YSTZ). According to their comments, we have addressed each concern raised by the reviewers in a point-by-point manner.

---

### Decision · Program_Chairs · 2024-09-25

**Decision:**

Accept (poster)

**Comment:**

The paper introduces a clean-to-clean reconstruction strategy to train DiT, which contrast with the noisy-to-noisy reconstruction that is performed in sota models. This contributes to the learning of more contextual information that benefit the reconstruction quality. Next to that, the authors propose an architecture with a 2-branch (complementary) DiT decoder to address the risk of mode collapse during learning. The novelty of the approach is evident and also supported by a theoretical analysis that strengthens the empirical results.
The experimental analysis is comprehensive, comparing performance results with existing methods (outperforming them on several tests) and performing ablations. The rebuttal has addressed the concerns of the reviewers, and the extra results and analyses provided in the author-reviewer discussions complement the already good results reported in the paper and appendices. In my opinion, they can be added to the supplementary material and referred at in the main paper.